# Spatial reasoning as Object Graph Energy Minimization

## Abstract

We propose a model that maps spatial rearrangement instructions to goal scene configurations via gradient descent on a set of relational energy functions over object 2D overhead locations, one per spatial predicate in the instruction. Energy based models over object locations are trained from a handful of examples of object arrangements annotated with the corresponding spatial predicates. Predicates can be binary (e.g., left, right, etc.) or multi-ary (e.g., circles, lines, etc.). A language parser maps language instructions to the corresponding set of EBMs, and a visual-language model grounds its arguments on relevant objects in the visual scene. Energy minimization on the joint set of energies iteratively updates the object locations to generate goal configuration. Then, low-level policies relocate objects to the inferred goal locations. Our framework shows many forms of strong generalization: (i) joint energy minimization handles zero-shot complex predicate compositions while each EBM is trained only from single predicate instructions, (ii) the model can execute instructions zero-shot, without a need for paired instruction-action training, (iii) instructions can mention novel objects and attributes at test time thanks to the pre-training of the visual language grounding model from large scale passive datasets. We test the model in established instruction-guided manipulation benchmarks, as well as a benchmark of compositional instructions we introduce in this work. We show large improvements over state-of-the-art end-to-end language to action policies and planning in large language models, especially for long instructions and multi-ary spatial concepts.

## 1 Introduction

Rearranging objects to semantically meaningful configurations has many practical applications for domestic and warehouse robots (Cakmak & Takayama, 2013). In this work, we focus on the problem of semantic rearrangement depicted in Figure 1. The input is a visual scene and a language instruction. The robot is tasked with rearranging the objects to their instructed configurations.

**End-to-end language to action mapping** Many works in semantic scene rearrangement have made progress by mapping language instructions directly to actions (Shridhar et al., 2021; Liu et al., 2021c; Janner et al., 2018; Bisk et al., 2017) or object locations (Mees et al., 2020; Gubbi et al., 2020; Stengel-Eskin et al., 2022). Many recent end-to-end language to action or language to object locations approaches use transformer architectures to fuse visual, language and action streams (Shridhar et al., 2022; Liu et al., 2021c; Pashevich et al., 2021). Despite their generality and their impressive results within the training distribution, these methods typically do not show generalization to different task configurations, for example, longer instructions, new objects present in the scene, novel backgrounds, or combinations of the above (Liu et al., 2021c; Shridhar et al., 2021). Furthermore, these methods cannot easily determine when the task has been completed and they should terminate (Shridhar et al., 2021; Liang et al., 2022) since they do not model explicitly the goal scene configuration to be achieved.

**Symbolic planners and learning to plan** To handle the challenges of reactive mapping of language to actions, some methods use look-ahead search to infer a sequence of actions or object rearrangements that eventually achieves the goal implied by the instruction (Prabhudesai et al., 2019). Look-ahead search for scene re-arrangement requires dynamics models that can handle complex

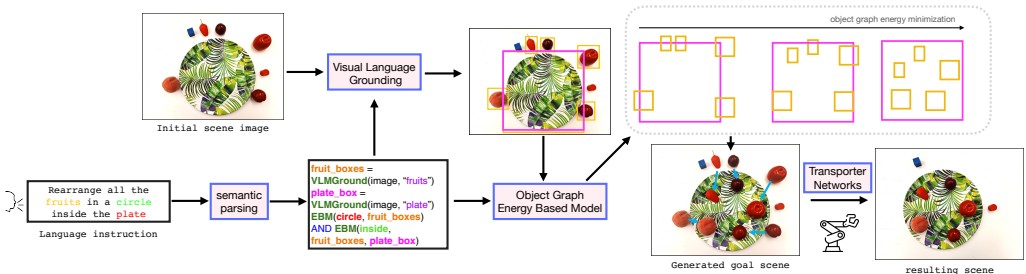

Figure 1: **Scene rearrangement as object graph energy minimization.**

multi-object scenes and generalize across varying number of objects and their configurations, which is an open area of research (LeCun, 2022). Latest learning-based planners (Hafner et al., 2019; 2020; Schrittwieser et al., 2019; Micheli et al., 2022) have not shown such generalization abilities (Goyal et al., 2021; Hamrick et al., 2020). As a result, planning for scene re-arrangement is dominated by symbolic planners, such as Planning Domain Definition Language (PDDL) planners (Migimatsu & Bohg, 2019; Kaelbling & Lozano-Pérez, 2011; Toussaint, 2015; Lyu et al., 2018). Symbolic planners assume that each state of the world, scene final goal states, and intermediate subgoal states can be sufficiently represented in a logical form, using language predicates that describe object spatial relations. These methods rely on manually specified symbolic transition rules and planning domains. They use object detectors for state estimation (Kase et al., 2020) and interface with low-level controllers for object manipulation. Many recent methods use learning to guide the combinatorial symbolic search by predicting directly semantic subgoals, conditioned on the instruction and the visual scene (Xu et al., 2019; Zhu et al., 2020; Drieß et al., 2020). Symbolic state bottlenecks used in these methods usually limits them to modelling pairwise spatial predicates. Spatial configurations that involve multi-ary relations between objects, such as circles, lines, squares, etc., are not easy to achieve because the long sequence of actions needed to put together such a shape configuration is outsourced to the low-level policy since intermediate states cannot be easily described in symbolic form (Figure-1). Under such long action horizon, the low-level policy often fails.

**Large language models for spatial planning** Recent methods wish to capitalize on knowledge of large language models (LLMs) (Brown et al., 2020; Liu et al., 2021b) for spatial planning (Huang et al., 2022a) and instruction following (Huang et al., 2022b; Ahn et al., 2022). Works of (Huang et al., 2022a) showed that LLMs can predict relevant language subgoals directly from the instruction and a symbolic scene description when conditioned on appropriate prompts, without any additional training (Huang et al., 2022b; Ahn et al., 2022; Huang et al., 2022a). The scene description is provided by an oracle (Huang et al., 2022b) or by object detectors (Huang et al., 2022b; Chen et al., 2022) as a list of objects. Predicted language subgoals, e.g., *"pick up the can"* are directly fed to low-level policies, or engineered controllers (Huang et al., 2022b). Work of (Liang et al., 2022) builds upon LLMs for code generation and predicts actions in the form of policy programs. It capitalizes on knowledge of LLMs for code generation regarding various functions necessary for semantic parsing of diverse language instructions. For the input programmatic scene description it uses lists of objects alongside their spatial box coordinates, missing from previous works. The language model predicts the actions in terms of low level program functions that interfaces with low level manipulation skills and object detection routines. The hope is that with appropriate prompting the LLM will draw from its vast implicit knowledge of code functions associated with natural language, provided in the comments of programs it has seen, to generate programmatic code that solves novel task instances of scene rearrangement. As highlighted in both (Liang et al., 2022; Huang et al., 2022b), prompting dramatically affects the model's performance and its ability to predict reasonable subgoals, and programmatic code, respectively. In this paper, we question whether the language space is the most efficient means to reason about objects and their spatial arrangements.

**Spatial reasoning as graph energy minimization** In this paper, we introduce Spatial Planning as multi Graph Energy Minimization (SPGEM), a framework for spatial reasoning for instruction following via compositional energy minimization over sub-graphs of object entities, one per spatial predicate in the instruction, as shown in Figure 1. We represent each spatial "predicate" as a binary or

multi-ari energy based model (EBM) over a graph of object spatial coordinates (LeCun et al., 2006). Predicates and corresponding EBMs can be binary (for concepts such as left, right, front, behind) or multi-ari (for circles, lines etc.). We train EBMs from examples of visual scenes paired with language descriptions of spatial relations, to predict gradients for object locations updates starting from randomly sampled object arrangements through Langevin dynamics minimization (Du et al., 2020b). Given a language instruction, e.g., *"put all the small objects in a circle inside the plate"*, a neural semantic parser maps the instruction to the set of predicates and corresponding EBMs, and grounds the object arguments for each EBM using visual language grounding models (Kamath et al., 2021; Jain et al., 2022). Gradient descent on the sum of energies with respect to the object coordinates provides the final object configurations that best satisfy the spatial constraints. Given the predicted locations of objects, we use pick and place visually grounded short term polices (Zeng et al., 2020) to move the objects to their inferred locations.

We show SPGEM generalizes zero shot to complex predicate compositions, e.g., "put all small objects in a circle in the plate" when trained from single relation predicate examples, e.g., " A inside B" or "a circle of A", as shown in Figure 1. Contrary to LLMs, SPGEM has been trained with a handful of visually grounded examples of spatial predicates. Contrary to symbolic planners, energy minimization "searches over" object placements while traversing non-symbolically meaningful scene configurations, and without manual engineering of the transition operators. Contrary to end-to-end language to action models, SPGEM infers a spatial abstraction of the goal scene that helps it to better evaluate whether the task has been completed. By representing EBMs over object locations as opposed to object visual pixel appearances (Du et al., 2020a; Liu et al., 2021a) the model can generalize better to zero-shot spatial concepts to novel objects. We build upon recent advances on training energy based models (Mordatch, 2018; Du et al., 2020b). To the best of our knowledge, this is the first work that uses EBMs for scene re-arrangement of complex language instructions.

We test SPGEM in scene re-arrangement of table top environments in simulation benchmarks of CLIPort (Shridhar et al., 2021) as well as in a new simulation benchmark we contribute with more compositional instructions. We curate multiple train and test splits to test out-of-distribution generalization with respect to i) longer instructions with more predicates, ii) novel object colours iii) novel objects iv) novel background colours. We compare our method with SOTA language to action models (Shridhar et al., 2021) as well as LLM planners (Huang et al., 2022b). We ablate each component of our model to isolate contributions of perception, language parsing, goal generation and low-level policy modules. Our model outperforms the baselines, especially in complicated instructions.

In summary, out contributions are: **(i)** An energy-based object-centric framework for language-guided goal scene generation. **(ii)** A modular system for language-guided scene re-arrangement, that uses semantic parsers, referential language grounding models, EBMs for scene generation and low-level vision-based policies for instruction following. **(iii)** Contribution of an extensive scene re-arrangement benchmark in simulation with compositional language instructions and shapes. **(iv)** Comparisons against SOTA end-to-end language to action models and LLMs, and extensive discussions of the pros and cons of each. We have included our code in the supplementary file. We will make our code and datasets publicly available upon publication.

## 2 RELATED WORK

**Instructing robots through language** Language robot instruction has a long history. Here, we focus our literature review on the task of spatial scene re-arrangement, and we refer the reader to the recent reviews of (Tellex et al., 2020; Luketina et al., 2019) for coverage of broader background work. Language is a natural means for communicating goals and can easily describe compositions of actions and arrangements seen at training time (Akakzia et al., 2021; Colas et al., 2020; 2022). Language is a much-preferred means of communicating a task than supplying goal images (Nair et al., 2018; Pong et al., 2019; Seita et al., 2021; Wu et al., 2022).We distinguish the following categories in the literature for robot instruction for scene re-arrangement: (i) End-to-end mapping of language to actions or language to object locations (Lynch & Sermanet, 2020; Shridhar et al., 2021; Liu et al., 2021c; Stengel-Eskin et al., 2022). (ii) Semantic parsers that map a language command and a visual scene description to programs over perceptual and control primitive functions (Karamcheti et al., 2020; Andreas et al., 2015; Walker et al., 2019; Matuszek et al., 2012; Wang et al., 2020; Srinet et al., 2020). These parsers require annotations of instructions with corresponding programs,

which are typically hard to collect at scale, despite recent progress on this topic through paraphrasing of instructions (Wang et al., 2015; Berant & Liang, 2014; Wu et al., 2021). Our work proposes a semantic parsing framework that instead of predicting action directly predicts EBMs constraints. We show such representation's ability to adapt few shots to complex spatial compositions. (iii) Symbolic planners that rely on predefined symbolic rules and known dynamic models, and infer discrete task plans given an instruction with logic search (Kaelbling & Lozano-Pérez, 2011; Garrett et al., 2018). These methods can generalize to arbitrary goals defined in a problem domain but necessitate the manual design of the planning domain, its predicates, symbolic rules and transition operators. (iii) Large language models (LLMs) that map instructions to language subgoals (Zhu et al., 2020; Xu et al., 2019; Huang et al., 2022a;b) or program policies (Liang et al., 2022). Subgoals predicted interface with low level short-term policies or skill controllers. LLMs trained from Internet-scale text have showed impressive zero-shot reasoning capabilities for a variety of downstream language tasks (Brown et al., 2020) when prompted appropriately, without any weight finetuning (Wei et al., 2022; Liu et al., 2021b). Researchers in robot learning have repurposed language models for the task of the instruction following (Chen et al., 2022; Lin et al., 2022). Works of (Huang et al., 2022a;b) have shown that LLMs can map instructions to a reasonable sequence of subgoals when prompted with analogous text action plans open loop or closed loop during instruction execution. The scene description is usually provided in a symbolic form as a list of objects present and their relations, predicted from open vocabulary detectors (Kamath et al., 2021). Recent works of (Liang et al., 2022; Lin et al., 2022) have also fed as input overhead pixel coordinated of objects to inform the LLM's predictions. The prompts for these methods need to be engineered per family of tasks. It is yet to be shown how the composition of spatial concept functions can emerge in this way.

**Language Conditioned Scene Generation.** A large body of work has explored scene generation conditioned on text descriptions (Johnson et al., 2018; Ramesh et al., 2021; Saharia et al., 2022; Yu et al., 2022; Mansimov et al., 2015). In this work, we seek to make scene generation useful as goal imagination for robot spatial reasoning and instruction following. Instead of generating pixel-accurate images, we generate object configurations by abstracting the appearance of object entities. We show this abstraction suffices for a great number of diverse scene rearrangement tasks.

**Energy based models** Our work is related to existing work on energy-based models (Grathwohl et al., 2019; Mordatch, 2018; Liu et al., 2021a; Du et al., 2020a;b; 2019). Most similar to our work is that of (Mordatch, 2018) which proposes a framework for learning to generate and detect spatial concepts with EBMs on images with dots, and (Du et al., 2020a; Liu et al., 2021a) which demonstrated composability of image-centric EBMs for RGB face and CLEVR image generation. Here we demonstrate zero-shot composability of graph-centric EBMs, their applicability on spatial reasoning, and their utility for vision-based instruction following for scene re-arrangement.

**Constraint Guided Layout Optimization**: Automatic optimization for object re-arrangement has been studied outside the field of robotics. (Yu et al., 2011) and (Merrell et al., 2011) use few user-annotated examples of scenes to adapt the hyperparameters of task-specific cost functions, which are then minimized using standard optimization algorithms (hill climbing and/or simulated annealing). To learn those hyperparameters from data, these approaches fit statistical models, e.g. Mixtures of Gaussian, to the given samples. (Fisher et al., 2012) further employ such optimization constraints into an interactive environment, where the user can provide an initial layout and the algorithm suggests improvements. All these approaches require expert knowledge to manually design rules and cost function, namely (Yu et al., 2011) identifies seven and (Fisher et al., 2012) eleven expert-suggested criteria for successful re-arrangement. Since they are hand-crafted, these methods do not generalize beyond the domain of furniture arrangement. In contrast, energy optimization is purely data-driven and domain-agnostic: a neural network scores layouts, assigning high energy to those that do not satisfy the (implicit) constraints and low energy to those who do, essentially modeling the underlying distribution of valid layouts.

## 3  METHOD

SPGEM's architecture is shown in Figure 1. We feed an RGB-D image of the scene and a language instruction. A semantic parser maps the instruction to a set of predicate EBMs and their arguments. A visual grounding model grounds the arguments of each EBM to object entities in the scene. Then, the goal configuration is predicted in a symbolic space of bounding boxes by minimizing the to-

tal energy wrt object locations. Lastly, short-term vision-based pick-and-place policies move the objects to their goal locations. We describe each component in more detail in the following.

**A library of graph energy-based models for spatial concepts**    In our work, a spatial predicate is represented by an EBM that takes as input $x$ the set of objects that participate in the spatial predicate and maps them to a scalar energy value $E_\theta(x)$. An EBM defines a distribution over configurations $x$ that satisfy its concept through the Boltzmann distribution $p_\theta(x) \propto e^{-E_\theta(x)}$. Low-energy graph entity configurations imply satisfaction of the language concept and have high probability. An example of the spatial concept can be generated by optimizing for a low-energy configuration through gradient descent on (part of) the input $x$. We represent each object entity by its 2D overhead centroid coordinates and box size. During gradient descent, we only update the center coordinates and leave box sizes fixed. We consider both binary spatial concepts (*in*, *left of*, *right of*, *in front of*, *behind of*) as well as multi-ary spatial concepts (*circle*, *line*).

During training, we sample configurations from $p_\theta$, by starting from an initial configuration $x^0$ and refining it using Langevin Dynamics (Welling & Teh, 2011):

$$x^{k+1} = x^k - \lambda \nabla_x E_\theta(x^k) + \epsilon^k z^k, \tag{1}$$

where $z^k$ is random noise, $\lambda$ is an update rate hyperparameter and $\epsilon^k$ is a time-dependent hyperparameter that monotonically decreases as $k$ increases. The role of $z^k$ and decreasing $\epsilon^k$ is to induce noise in optimization and promote exploration, similar to Simulated Annealing (Kirkpatrick, 1984) After $K$ iterations, we obtain $x^- = x^K$. During training, we learn the parameters $\theta$ of our EBM using a contrastive divergence loss that pushes the energy of sampled $x^-$ to be higher than the energy of ground-truth $x^+$:

$$\mathcal{L} = \mathbb{E}_{x^+ \sim p_D} E_\theta(x^+) - \mathbb{E}_{x^- \sim p_\theta} E_\theta(x^-),$$

where $x^+$ a sample for the data distribution $p_D$ and $x^-$ a sample drawn from the learned distribution $p_\theta$. Compositions of concepts can be created by simply summing energies of constituent concepts, e.g., *"a circle of cubes inside the plate"*, as shown in Figure 1. For detailed implementation and architectural figures of our energy-based models, please see Section-6.1.2.

**Semantic parsing of instructions into spatial concepts graphs and their arguments**    Our parser maps language instructions to instantiations of energy-based models and their arguments. Examples of instructions and paired programs are shown in Figure 1. Our parser is a Sequence-to-Tree model (Dong & Lapata, 2016) with a copying mechanism (Gu et al., 2016) which allows it to handle a larger vocabulary than the one seen during training. Specifically, the input utterance is first encoded using a pre-trained RoBERTa encoder (Liu et al., 2019), giving a sequence of contextualized word embeddings and a global representation of the full utterance. Then, a decoder is iteratively employed to either i) decode an operation (`EBM` or `VLMGround`), ii) condition on this operation to decode or copy the arguments for this operation, iii) add one (or more) child node(s).

**Referential visual language grounding**    We ground noun phrases predicted by our parser with an off-the-shelf language vision grounding model (Jain et al., 2022). The input is the noun phrase, e.g., "the blue cube" and the image, and the output is the boxes of all object instances that satisfy the noun phrase. We finetune the publicly available code of (Jain et al., 2022) to our training data.

**Short-term vision-based manipulation skills**    Our short-term manipulation skills build upon transporter networks (Zeng et al., 2020). Transporter Networks take as input an overhead RGB-D image registered at the overhead bird view and predict two robot gripper poses: i) a pick pose and ii) a pick-conditioned placement pose. Transporters can model any behaviour that can be effectively represented as two consecutive poses for the robot gripper, such as pushing, sweeping, rearranging ropes, folding, and so on. For more details on transporters please refer to (Zeng et al., 2020). We modify Transporter Networks to be locally conditioned. Specifically, we crop an image patch using the predicted pick and place locations from our visual grounding and EBM modules respectively. In this way, the low-level policies know roughly what to pick and where to place it, and only locally optimize over the best pick location (within an object of interest) or placement location, at a particular part of the scene, respectively.

**Close Loop Feedback**: Our model generates its own goal and then goes on to execute it. Once it executes it, we re-detect all relevant objects using our VLM-grounder module and check if they

are close to their predicted goal location. Concretely, if the re-detected object's bounding box and predicted goal bounding box has a positive Intersection over Union (IoU) we consider the goal to be successfully executed. If we fail to reach the goal, we retry the instruction again starting from the present scene configuration. For more details, please refer to Section-6.2.2.

# 4 EXPERIMENTS

We test SPGEM in its ability to follow instructions for re-arranging tabletop scenes in simulation and in the real-world. We compare our model against large language model planners (LLM) (Huang et al., 2022b) and end-to-end language to action models (Shridhar et al., 2021).

Our experiments aim to answer the following questions: **1)** How does the proposed set energy minimization over object configurations compare with LLM planners in predicting object arrangements from spatial instructions? How much this ability scales with the length of the instruction? **2)** How does SPGEM compare with state-of-the-art end-to-end instruction to action models across varying instruction lengths? **3)** How do different modules of our framework contribute to performance?

**Baselines** We compare SPGEM against the following baselines:

**(i)** CLIPort (Shridhar et al., 2021), a model that takes as input an overhead RGB-D image and an instruction and uses pretrained CLIP encoders to featurize the instruction and RGB image, respectively, then fuses with depth features to predict pick and place actions using the parametrization of Transporter Networks (Zeng et al., 2020). Its aim is to capitalize on language vision associations already learnt by the CLIP encoders. We use the publicly available code of (Shridhar et al., 2021).

**(ii)** LLMplanner (Huang et al., 2022b) is an instruction following scene rearrangement model that uses Instruct-GPT to predict sequential low-level subgoals in language form, e.g., *"pick the red cube and place it to the right of the blue bowl"*. These predictions interface with vision and language-conditioned short-term policies, such as CLIPort. Scene state description is provided as a list of objects in the scene. LLMplanner does not finetune the LLM but instead uses appropriate prompts.

**Benchmarks** We test our models and baselines for scene re-arrangement in the following benchmarks: (i) The four tasks of put-block-in-bowls, pack-google objects-seq, pack-google objects-group, and assemble-kits-seq from CLIPort benchmark (Shridhar et al., 2021). (ii) Scene rearrangement benchmarks that we designed programatically in PyBullet simulator (more details in Section-6.5). These include: a) *Spatial Relations benchmark*, which contains single pick 'n' place instructions with referential expressions in cluttered scenes with distractors, e.g., *"Put the cyan cube above the red cylinder"* b) *Compositional-one step* scene rearrange benchmark, which contains compositional instructions with referential expressions that require one object to be re-located to a particular placement location, e.g., *"put the red bowl to the right of the yellow cube and left of the red cylinder and above blue cylinder"*. c) *Compositional-group* scene rearrange benchmark, which contains compositional instructions with referential expressions that require multiple objects to be re-located, e.g., *"put the grey bowl above the brown cylinder and put the yellow cube to the right of the blue ring and put the red bowl above the blue ring and put the blue ring below the grey bowl."*. d) *Shapes benchmarks*, which contain instructions for making multi-entity shapes, such as circles and lines, e.g., *"rearrange all red cubes in a circle"*

**Evaluation Metrics** We use two evaluation metrics: (i) **Task Progress** was proposed in the Ravens benchmark (Zeng et al., 2020) and represents the percentage of the referred objects placed in their goal location, e.g., 4/5 = 80.0% for rearranging 4 out of 5 objects specified in the instruction. (ii) **Task completion** rewards the model only if the full rearrangement is complete. For the introduced benchmarks we have oracle reward functions that evaluate whether the task constraints are satisfied.

## 4.1 SCENE RE-ARRANGEMENT UNDER ORACLE PERCEPTION AND LOW-LEVEL CONTROL

Our goal is to compare spatial reasoning in a language space vs in an abstract visual space for predicting instruction-guided subgoals. We compare our model with LLMplanner. In this section, we assume oracle perception, that is, oracle referential grounding for both LLMplanner and SPGEM, as well as oracle low-level control policies. We carry out inferred instructions from LLMplanner using

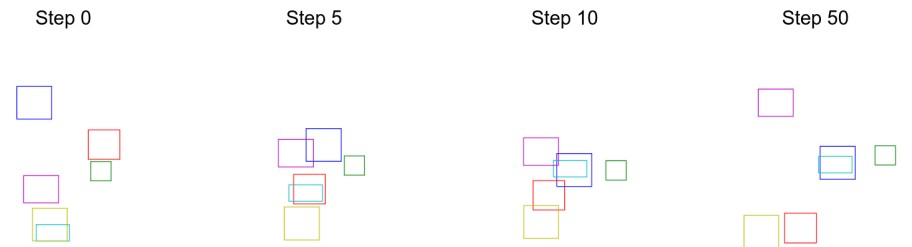

Step 0      Step 5      Step 10      Step 50

*"Place the blue box left of the green, right of the red box and above the yellow box. Put the cyan box inside the blue box. Move the red box below the magenta box."*

Figure 2: **Spatial planning via composable concept graph energy minimization.** The EBMs handle complex instructions as multiple steps of joint energy minimization. Step 0 indicates the starting point and step 50 the result after 50 gradient steps. Our model "imagines" the goal scene in a pixel-abstract but spatially-aware manner. Note that the optimization is not aware of physical constraints: in intermediate time steps the objects can overlap, e.g. steps 5 and 10. However, this is a planning procedure and the intermediate steps, where the energy is high, are not executed.

oracle controllers that relocate the right objects to locations in the scene that satisfy the predicted instruction while not occupying other objects. Note however though that SPGEM relies on simple pick and place policies that are not language conditioned, while the LLMplanner relies on language-conditioned policies, thus, the oracle control assumption may be more unrealistic in the latter case. We forego this difference for the sake of comparison.

We show quantitative results of SPGEM and LLMplanner on compositional-one-step and group benchmarks in Table 1. Our model outperforms LLMplanner, and the performance gap is larger in more complex instructions. To elucidate why an abstract visual space may be preferable for spatial planning, we visualize steps of energy minimization for different instructions in Figure 2 and we visualize steps of the execution of the InstructGPT prompted by us to the best of our capability in Figure 3. As you can see, SPGEM trained on single predicate scenes shows remarkable composability to much more predicates. Language planning on the other hand suffers from the ambiguity of translating geometric concepts to language and vice versa: step-by-step execution of language subgoals does not suffice for the composition of the two subgoals to emerge. (Figure 3).

| Method | Composition-one-step | | Composition-group | |
| --- | --- | --- | --- | --- |
| | Task Progress | Task Completion | Task Progress | Task Completion |
| LLMplanner oracle | 82.0 | 59.0 | 75.3 | 29.0 |
| SPGEM oracle | **90.8** | **76.0** | **88.7** | **62.0** |

Table 1: **Evaluation of SPGEM and LLMplanner with oracle perception and oracle low-level execution policies** on compositional spatial arrangement tasks.

## 4.2 Spatial scene re-arrangement

**Simulation:** In this section, we compare our model and baselines in the task of instruction-guided scene re-arrangement. We show quantitative results in Table 2 for the benchmarks of Composition-one-step and Composition-group. We compare our model with CLIPort model trained on atomic spatial relations and zero-shot evaluated on compositional benchmarks. We further fine-tune the CLIPort on demos from the compositional benchmark. Under all different settings, we vastly outperform all variants of CLIPort and LLMplanner. We also observe that closed-loop feedback boosts our performance further. Specifically for compositional tasks, our model does not need any retraining because the energy functions are composable, meaning that we can jointly optimize over an arbitrary number of constraints by simply summing the different energy terms, however, we need few synthetic labels from our grammar for the language parser. We also show results on Spatial Relations and shapes benchmark in Table 7 where we vastly outperform all our baselines. We also show results for cliport benchmark and detailed generalization tests in the Section-6.2.1.

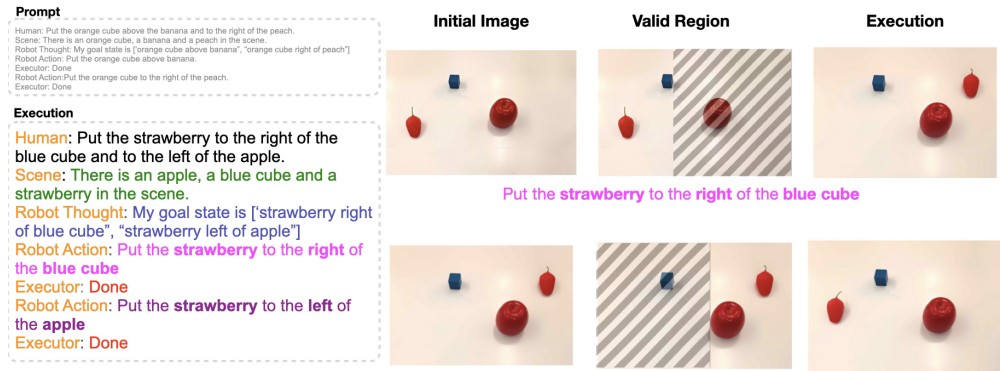

Figure 3: **Spatial planning via Large Language Models (LLMs).** LLM Planners describe their plans in language actions supported by the executor policy. However, this can be limiting. Here, the policy supports four atomic relations i.e. left, right, above and below. LLM instructs the executor to put the strawberry to the right of the cube which the executor can place anywhere in the shaded area. However, in the next instruction of putting the strawberry to the left, the executor violated the first constraint and ended up in a similar initial configuration. Note that while "middle" is a valid English relation that can convey both constraints jointly here, however, a) it is not supported by the executor and b) words do not exist for every configuration especially when the joint constraint increases.

| | Composition-one-step seen-colors | | Composition-one-step unseen-colors | | Composition-group seen-colors | | Composition-group unseen-colors | |
|---|---|---|---|---|---|---|---|---|
| Method | 10 | 100 | 10 | 100 | 10 | 100 | 10 | 100 |
| Initial (no movement) | 0.0 | 0.0 | 0.0 | 0.0 | 31.7 | 31.7 | 31.8 | 31.8 |
| CLIPort (zero-shot) | 9.0 | 12.0 | 7.0 | 12.0 | 37.4 | 37.5 | 32.6 | 38.4 |
| CLIPort | 13.0 | 15.0 | 14.0 | 9.0 | 38.2 | 38.5 | 34.7 | 40.9 |
| LLMplanner | 51.2 | 53.2 | 49.4 | 53.5 | 38.6 | 39.0 | 37.1 | 39.0 |
| Ours (zero-shot) | 90.0 | 91.0 | 92.7 | 90.3 | 77.2 | 77.4 | 77.7 | 78.4 |
| Ours (zero-shot + feedback) | **91.6** | **92.0** | **92.9** | **91.4** | **80.8** | **81.6** | **81.1** | **82.4** |

Table 2: **Evaluation of SPGEM and CLIPort on compositional tasks** SPGEM is trained only on atomic relations and tested zero-shot on tasks with compositions of spatial relations. Composition-one-step involves moving only one object to satisfy all constraints specified by the language. Compositional group task involves moving all objects to satisfy the task constraints. Some language constraints are satisfied already in the initial configuration, and the Initial model captures that.

**Real-World:** We test our model in the real world using a 7-DoF Franka Emika robot equipped with a parallel jaw gripper. We do not do any real world finetuning. Our test set contains 10 language-guided table top manipulation tasks per setting (composition-one-step, composition-group, circles, lines). We show quantitative results in Table 5. We observe that SPGEM generalizes to the real world thanks to the open vocabulary detector trained in the real world, and the object abstractions in the concept EBMs and low-level policy modules.

## 4.3 ABLATIONS

We do ablations and error analysis of our model in Table-4. First, we remove the goal generation from SPGEM by conditioning the place network on the language input instead of the EBM-generated goal image while keeping the pick network and object grounders identical. We observe a drop of 35.1% in accuracy underscoring the importance of goal generation. We then remove our executor policy and instead randomly select pick and place locations inside the bounding box of the relevant object. This results in a drop of 16% showing the importance of robust low-level policies. We do not remove the grounder and parser since they are necessary for goal generation. We then experiment with oracle visual language grounder that perfectly detects the objects mentioned in the sentence and we observe a gain of 5.1%. We posit that this difference would be even higher with real-world scenes. We observe that language parsing in our domain is close to perfect, hence we skip that. We finally evaluate with perfect grounding, language parsing and low-level execution to

| Method | left-seen-colors | | left-unseen-colors | | right-seen-colors | | right-unseen-colors | |
|---|---|---|---|---|---|---|---|---|
| | 10 demos | 100 demos | 10 demos | 100 demos | 10 demos | 100 demos | 10 demos | 100 demos |
| CLIPort-multi | 13.0 | 44.0 | 9.0 | 33.0 | 29.0 | 43.0 | 28.0 | 44.0 |
| Ours | **95.0** | **95.0** | **93.0** | **94.0** | **89.0** | **92.0** | **93.0** | **96.0** |

| Method | above-seen-colors | | above-unseen-colors | | below-seen-colors | | below-unseen-colors | |
|---|---|---|---|---|---|---|---|---|
| | 10 | 100 | 10 | 100 | 10 | 100 | 10 | 100 |
| CLIPort-multi | 24.0 | 45.0 | 22.0 | 51.0 | 23.0 | 55.0 | 13.0 | 40.0 |
| Ours | **87.0** | **87.0** | **89.0** | **90.0** | **89.0** | **90.0** | **88.0** | **89.0** |

| Method | circle-seen-colors | | circle-unseen-colors | | line-seen-colors | | line-unseen-colors | |
|---|---|---|---|---|---|---|---|---|
| | 10 demos | 100 demos | 10 demos | 100 demos | 10 demos | 100 demos | 10 demos | 100 demos |
| CLIPort-multi | 34.1 | 61.5 | 31.2 | 55.6 | 48.6 | 88.2 | 48.6 | 88.5 |
| Ours | **91.3** | **91.5** | **90.2** | **91.2** | **98.1** | **99.0** | **98.4** | **99.4** |

Table 3: **Evaluation of SPGEM and CLIPort on spatial relations and shapes in simulation.**

| Model | Accuracy |
|---|---|
| SPGEM | 77.2 |
| SPGEM without goal generation | 42.1 |
| SPGEM w/o learnable policies | 61.2 |
| SPGEM with oracle language visual grounding | 82.3 |
| SPGEM with everything oracle except goal | 88.3 |

| Task | Performance |
|---|---|
| Composition-one-step | 85.6 |
| Composition-group | 75.8 |
| Circles | 94.0 |
| Lines | 90.0 |

Table 4: **Ablations of SPGEM in the benchmark compositions-group-seen-colors.**

Table 5: **Evaluation of SPGEM in the Real World**

test the error rate of our goal generator. We obtain 88.3% accuracy thus concluding that our goal generator fails in 11.7% cases. For more detailed error analysis, please refer to Secs. 6.2.3 and 6.2.4 of the Appendix.

## 4.4 LIMITATIONS

Our model presently has two important limitations. First, it predicts the goal object scene configuration but does not have any knowledge regarding temporal ordering constraints implied by physics. For example, our model can predict a stack of multiple objects on top of one another but cannot suggest which object needs to be moved first. Adding such temporal energy priors is a direct avenue for future work. Second, our EBMs are currently parametrized by object locations but many tasks, e.g, manipulation of articulated objects, fluids, deformable objects or granular materials, would require finer grain parametrization in both space and time. Furthermore, even for rigid objects, many tasks would require finer in-space parametrization, e.g., it would be useful to know a set of points in the perimeter of a plate as opposed to solely representing its centroid and box for accurately placing things inside it. For more details, please refer to Section-6.3.

## 5 CONCLUSION

We introduced SPGEM, a modular framework for instruction-guided scene re-arrangement that maps language to object scene configurations via compositional energy minimization over detected object graphs, one per spatial concept. We showed better generalization of our framework to predicate concept compositions, novel object combinations, and object scene arrangements never seen at training time. We test our model in diverse table top manipulation tasks in simulation and show improvements both in and out-of-distribution settings, as well as in low training data regimes, against state-of-the-art end-to-end language to action mapping models, and LLM-based instruction following methods. We contributed a new scene rearrangement benchmark for more compositional language instructions, which we will make publicly available, alongside our code, upon publication. Our work questions the efficiency of spatial reasoning in language space, and shows a handful of examples suffice to learn EBM concepts, and compose them over novel arrangements.

## REPRODUCIBILITY STATEMENT

To ensure the reproducibility of the empirical results, we submit a zip file with an anonymized version of our code as part of the supplementary file. Upon publication, we will make our full code, data and benchmark publicly available on GitHub.

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

# 6    APPENDIX

For visualizations and real robot execution videos, please visit our website: https://sites.google.com/view/spgem/

## 6.1    MODEL DETAILS

Please refer to Algorithm-1 for a simplified pseudo-code for our pipeline. We provide more details in the following sections.

---

**Algorithm 1** Pseudo code for SPGEM's pipeline

---

```
# Inputs:
# img: Input RGB image, depth: Associated depth image, lang: input task
    instruction

# Model
# VLM-Grounder: A language grounding modules which detects objects mentioned in
    the sentence
# language_parser: A Seq-to-Tree model which maps natural language to an
    executable program
# ebm_library: Library of EBMs consisting of atomic binary concepts like left,
    right, above, below and inside and n-ary concepts like circle and lines
# gc_transporter: low level pick-and-policy which executes the pick and place
    action conditioned on the goal from EBM

for img, depth, lang in dataloader: # load one task inputs
    program = language_parser(lang) # the parser spits out a program over the
        Grounder and EBMs

    intermediate_outputs = []
    for i, prog in enumerate(program): # iterate over program from parser
        if prog['module'] == 'filter':
            object_desc = prog['concept'] # description of object to detect for eg: "
                the left green cube"
            object_box = VLMGround(img, object_desc)
            intermediate_outputs.append(object_box)
        elif prog['module'] == 'EBMs':
            subjs = prog['subj'] # list of lists, rel boxes, e. g. [[1], [0, 9]],
                indexes into intermediate_outputs
            objs = prog['obj'] # list of lists, ref boxes, e. g. [[2], []], indexes
                into intermediate_outputs
            rels = prog['rels'] # list of str, relation names, e. g. ["right", "
                circle"]
            picks = prog['picks'] # list of objects that need to be picked, indexed
                into intermediate outputs

            goal_boxes = run_ebm(subjs, objs, rels, intermediate_outputs) # runs the
                langevin dynamics on the required EBMs with appropriate boxes

            pick_boxes = intermediate_outputs[pick_boxes]
        else:
            assert False

    # predict pick and place locations conditioning on bounding boxes of pick and
        place goals
    pick_locs, place_locs = gc_transporter(img, depth, pick_boxes, goal_boxes)

    # robot controller actually moving the end effector to execute the task
    low_level_controller(pick_locs, place_locs)
```

---

### 6.1.1    TRAINING A SEMANTIC PARSER WITH SYNTHETICALLY GENERATED UTTERANCE-PROGRAM PAIRS

**Domain Specific Language for grounding and Generation**    We design a Domain-Specific Language (DSL) which extends the DSL of NS-CL (Mao et al., 2019) (designed for visual question answering in CLEVR (Johnson et al., 2017)) to further predict scene generations, e.g. *"put all brown shoes in the green box"*. Detailed description of our DSL can be found in Table 12.

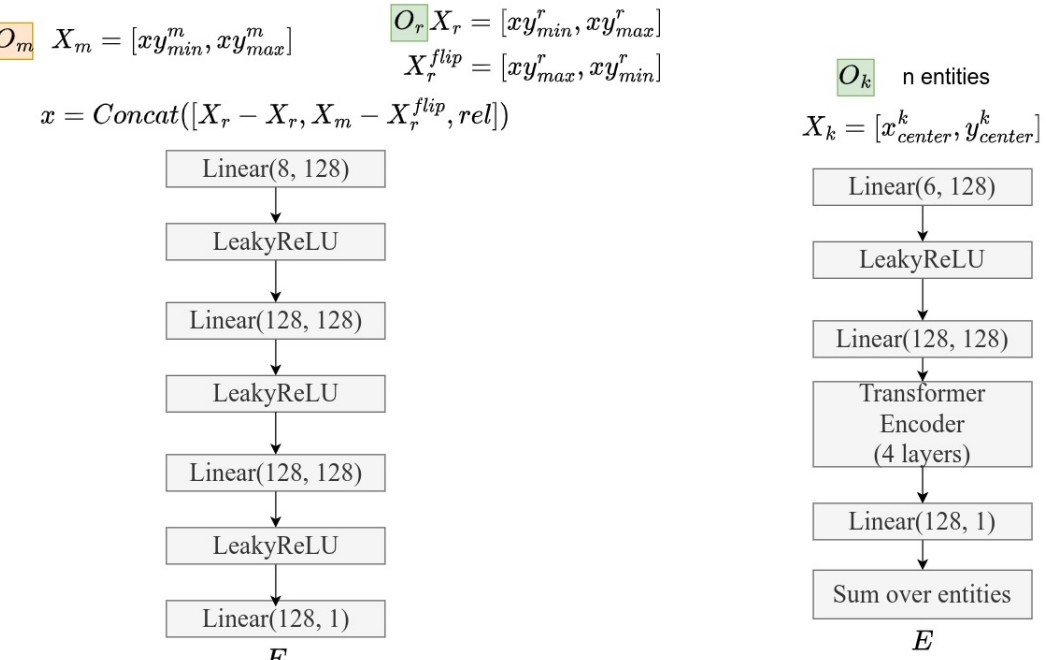

Figure 4: *Left*: Architecture of the EBM used for binary concepts such as "right of". The inputs are a "moving" point $O_m$ and a "reference point" $O_r$ and the output is the energy of their relative placement. *Right*: Architecture of the EBM used for multi-ary concepts such as "circle". The input is a set of $n$ entities $O_k, k = 1, \ldots, n$. The output is the energy of this set of entities.

### 6.1.2 EBMs

**Implementation Details:** We implement two EBMs, namely a BinaryEBM and a MultiAryEBM, for binary (e.g. "right of") and multi-ary (e.g. "circle") concepts respectively. The BinaryEBM is used for all tasks that ask for moving objects with respect to some object of reference, e.g. as in the "Put" and "Pack" tasks of the CLIPort benchmark. We train a separate EBM for each basic concept, i.e. "left", "right", "inside", "above", "below", etc. A BinaryEBM takes as input two objects and learns to move the first object with respect to the second one. A MultiAryEBM receives a set of objects and re-arranges them to a shape. Again we build a separate EBM for each shape concept, e.g. "circle" and "line".

We implement the BinaryEBM as a relational network that operates on the relative locations between the objects (Figure 4). Then, a multi-layer perceptron (MLP) is used to map the featurized pair to a scalar value, which is the energy.

For the MultiAryEBM, we first featurize each object by applying a linear layer and a LeakyReLU activation on top of its absolute coordinates. Then, we feed the set of object features to a sequence of four Transformer encoder layers (Vaswani et al., 2017) for contextualization. The refined features are averaged into an 1D vector which is then mapped to a scalar energy using an MLP. The architecture is depicted in Figure **??**.

During training, we sample positive and negative examples and optimize for Equation 2. We additionally use the KL-loss and the L2 regularization proposed in (Du et al., 2020b). We refer the reader to that paper for more explanation on these loss terms.

**Extension to tasks with pose**: An important design choice is what parameters of the input we should be able to edit. We inject the prior knowledge that on our manipulation domain the objects move without deformations, so we fix their sizes and update only their positions. Our EBMs operate on boxes so that they can abstract relative placement without any need for object class or shape

information. However, EBMs can be easily extended to optimize other types of representations, such as pose.

We train an EBM, we call it PoseEBM, to optimize the 2D angle of a posed object wrt a point of reference. This is a common constraint in furniture re-arrangement, e.g. "the armchair should look towards the tv". For this task, we parameterize a posed object as a point (its centroid) and an angle (the orientation of its main axis). The PoseEBM takes as input one such object and a reference point. First, it computes the relative location of the posed object wrt to the reference point. Then, the relative location and the angle are concatenated to a vector of three elements. This is fed to an MLP that outputs a scalar energy value. We sample positive examples of scenes where the posed object "looks towards" the point of reference, i.e. its main axis is parallel to the vector connecting its centroid and the reference point.

## 6.2 EXPERIMENTS

### 6.2.1 GENERALIZATION EXPERIMENTS

- **Generalization Benchmark / Generalization Tests**: We conduct controlled studies of our model's generalization across three axes: a) Novel Colors b) Novel background color of the table c) Novel Objects. In each of these settings, we only change one attribute (i.e. object color, background color or object instance) while keeping everything else constant.

  **Novel colors**: We train the models with ['blue', 'red', 'green', 'yellow', 'brown', 'gray', 'cyan'] colors and evaluate them with unseen colors: ['orange', 'purple', 'pink', 'white'].

  **Novel background colors**: All models are trained with black colored tables and evaluated with randomly sampled RGB color for each instruction.

  **Novel objects**: We train the models on ["ring", "cube", "cylinder", "bowl"] and evaluate them with ["triangle", "square", "plus", "diamond", "pentagon", "rectangle", "flower", "star", "circle", "hexagon", "heart"].

  We evaluate ours and cliport model trained on 10 or 100 demos on spatial reasoning benchmark tasks (involving single left, right, above, below) and composition benchmarks (composition-one-step, composition-group). The results are summarized in the table-6 and in the figure-5.

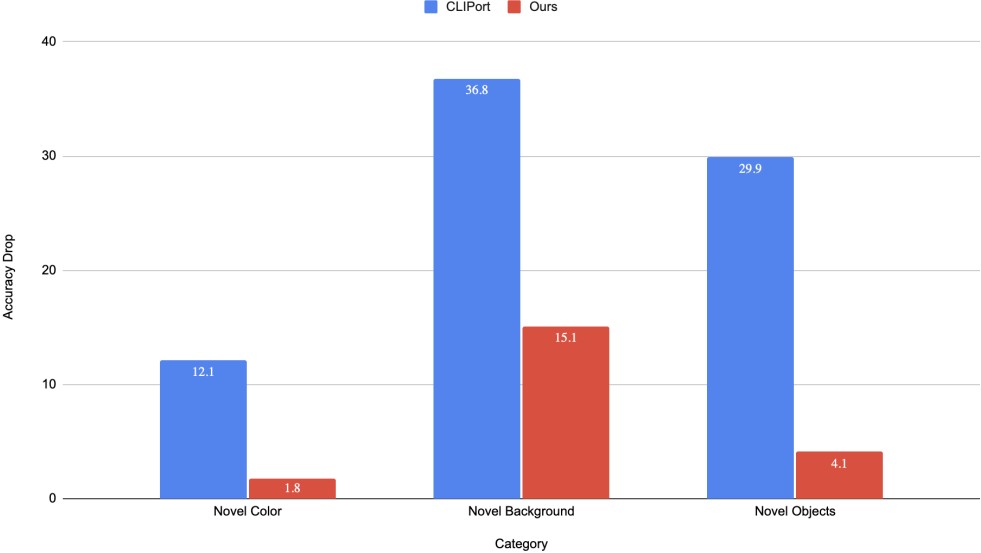

Figure 5: **Generalization comparison between CLIPort and Ours**: Average percentage drop by changing color of objects or backgroud color or introducing novel objects (lower is better).

| Attribute | Model | Spatial-seen | | Spatial-novel | | Composition-seen | | Composition-novel | |
|---|---|---|---|---|---|---|---|---|---|
| | | 10 demos | 100 demos | 10 demos | 100 demos | 10 demos | 100 demos | 10 demos | 100 demos |
| Color | CLIPort | 22.0 | 47.0 | 18.0 | 39.0 | 25.6 | 26.8 | 25.1 | 24.5 |
| | SPGEM (ours) | 90.0 | 91.0 | 87.0 | 85.0 | 83.6 | 84.2 | 86.5 | 84.0 |
| Background | CLIPort | 22.0 | 47.0 | 10.0 | 20.0 | 25.6 | 26.8 | 23.7 | 23.2 |
| | SPGEM (Ours) | 90.0 | 91.0 | 79.0 | 68.0 | 83.6 | 84.2 | 77.0 | 72.0 |
| Objects | CLIPort | 22.0 | 47.0 | 17.0 | 19.0 | 25.6 | 26.8 | 24.5 | 24.8 |
| | SPGEM (Ours) | 90.0 | 91.0 | 86.0 | 86.0 | 83.6 | 84.2 | 80.9 | 81.5 |

Table 6: **Generalization experiments of SPGEM and CLIPort in manipulation tasks in simulation.**

| Method | put-block-in-bowl seen-colors | | put-block-in-bowl unseen-colors | | packing-google-objects seq-seen-objects | | packing-google-objects seq-unseen-objects | |
|---|---|---|---|---|---|---|---|---|
| | 10 demos | 100 demos | 10 demos | 100 demos | 10 demos | 100 demos | 10 demos | 100 demos |
| CLIPort-multi | 31.0 | 82.1 | 4.8 | 17.6 | 34.8 | 54.7 | 27.2 | 56.4 |
| Ours | 84.3 | 93.8 | 89.0 | 95.3 | 86.8 | 94.8 | 88.0 | 92.9 |

| Method | packing-google-objects group-seen-objects | | packing-google-objects group-unseen-objects | | assembling-kits seq-seen-colors | | assembling-kits seq-unseen-colors | |
|---|---|---|---|---|---|---|---|---|
| | 10 | 100 | 10 | 100 | 10 | 100 | 10 | 100 |
| CLIPort-multi | 33.5 | 61.2 | 32.2 | 70.0 | 38.0 | 62.6 | 36.8 | 51.0 |
| Ours | 86.1 | 76.8 | 87.2 | 79.6 | 38.4 | 42.0 | 40.8 | 44.0 |

Table 7: **Evaluation of SPGEM and CLIPort on CLIPort benchmark in simulation.**

**Analysis**: We observe that on average on a) Novel Colors our model's performance dropped by 1.% compared to 12.1% for CLIPort b) on novel backgrounds our drop is 15.1% compared to 36.8% for CLIPort with most of our failures caused in transporter-based low-level policy c) on novel backgrounds our drop is 4.1% compared to 29.9% for CLIPort.

- **CLIPort's benchmark**: We also show generalization experiments across novel colours and novel objects across the established CLIPort benchmark. Please refer to the original CLIPort paper (Shridhar et al., 2021) for more details of the benchmark. Note that the original CLIPort benchmarks assume access to oracle success/failure information based on which it can retry the task for a fixed budget of steps or stop the execution if oracle confirms that the task is completed. We evaluate the CLIPort model without this oracle retry but still with oracle information of how many minimum steps it needs to take to complete the task. For example, if the task is "Place all strawberries in the bowl", it has access to the oracle information of the number of strawberries, say $N$, in the scene and it tries to pick and place the strawberries for $N$ steps. Our model, in contrast, fires a language grounding model to find the number of task-relevant entities in the scene, without assuming any oracle information. The *put-block-in-bowl* and *assembling-kits-seq* task tests generalization across colours while all remaining tasks test generalization across novel objects. In this benchmark as well we observe our model generalizes gracefully across novel colours and objects while CLIPort model can suffer significant performance hits. We also outperform CLIPort on all the tasks by a margin except the assembling task when trained on 100 demos. Most of the failure cases there are due to the language grounder getting confused between letters and letter holes.

- **Real World Experiments** In our real world experiments, we also use novel objects or novel object descriptions like "bananas", "strawberry", "small objects" which the model hasn't seen the simulation training.

Our model's generalization capabilities rely on an open-vocabulary detector and the fact that EBMs and transporter-based low-level execution policy operate on abstracted space in a modular fashion. While CLIPort models can also generalize to novel scenarios, thanks to CLIP, however, the action prediction and perception are completely entangled and hence even if CLIP manages to identify the right objects based on the language, it has trouble predicting the correct pick and place locations.

| Benchmark | Precision | Recall | Accuracy |
|---|---|---|---|
| Composition-group | 80.5 | 82.5 | 85.0 |
| Composition-one-step | 71.4 | 19.2 | 77.0 |

Table 8: **Performance of failure detection using our generated goal**

### 6.2.2 CLOSED LOOP FEEDBACK

We also experiment by adding closed-loop feedback to our model and retry the instruction if the feedback mechanism inferred a failure.

**Implementation**: Our model generates its own goal and then goes on to execute it. Once it executes it, we re-detect all relevant objects using our VLM-grounder module and check if they are close to their predicted goal location. Concretely, if the re-detected object's bounding box and predicted goal bounding box has a positive Intersection over Union (IoU) we consider the goal to be successfully executed. If we fail to reach the goal, we retry the instruction again starting from the present scene configuration (without resetting the environment, since it's a big assumption).

**Results**: As we show in the main paper, adding feedback roughly adds 3.5% performance boost in composition-group benchmark and 1% boost in composition-one-step benchmark. Retrying cannot always recover from the failure, however, it would be still important to know if the execution failed so that we can request help from a human. Towards this, we measured the failure detection capabilities of our feedback mechanism. Specifically, we measured the precision, recall and accuracy of the failure detection mechanism. The results are in Table-8. We observe that all metrics are high for composition-group benchmark. For composition-one-step, however, the recall is very low i.e. only 19.2%. This is because in this benchmark, we only need to move one object to complete the task and hence most failures are due to wrong goal generation. Thus, the failure classifier classifies those demos as success, because indeed the model managed to achieve its predicted goal and hence results in low recall. This motivates the use of external failure classifiers in conjunction with internal goal-checking classifiers. In contrast, composition-group benchmark is harder because it requires the model to move many objects and thus results in higher chances of robot failures. These failures can be better detected and fixed by our goal-checker classifier.

Also, note that while previous works like InnerMonologue (Huang et al., 2022a) showed huge improvements by adding closed-loop feedback, the improvements for us are smaller. This is because, by design, our model is more likely to reach its goal – since EBM generates a visual goal and then the low-level policy predicts a pick and place location **within** the predicted goal, it is very likely to satisfy its predicted goal. Indeed, in the composition-one-step, the model satisfies its goal in 93% cases and in 70% cases in the composition-group benchmark. In contrast, InnerMonologue does not have any built-in mechanisms for promoting goal-reaching behaviours and thus the difference in performance with additional goal-satisfying constraints is larger for them.

**Limitations**: There is a lot of potential in designing better goal checkers. As already discussed, we should incorporate external success classifiers like those used by prior literature (Huang et al., 2022a) in conjunction with goal-checking classifiers. Explicit goal generation allows then to check validity of goals directly even before actual robot execution (which makes it safer and less expensive). We leave this for future work. Another promising direction is to add object trackers in the feedback mechanism to keep track of the objects and detect if a failure happened. This is important, if for example, we have multiple objects that are visually similar and hence we would need to keep track of which object is supposed to go to which goal and retry if it failed to reach its goal.

### 6.2.3 ERROR ANALYSIS AND ROBUSTNESS TO NOISE

We conduct detailed error analysis of our model on composition-group benchmark and report results in table-9. We find that robot failures, which include collisions or failure to pick/place objects, result in 6.0% drop in accuracy. Goal generation adds 11.7% to the failure. Visual grounding leads to 5.1% errors while we find language parsing to be nearly perfect.

To test robustness of our modules from noisy outputs from parent modules, we add gaussian scaling and translation noise to predictions from grounding and goal generation modules. We add a scaled gaussian noise of varying percentage as a fraction of the size of the objects. The results are sum-

| Error Mode | Error Percentage |
|---|---|
| Robot failure | 6.0 |
| Goal Generation failure | 11.7 |
| Grounding failure | 5.1 |
| Language Parsing failure | 0.0 |

Table 9: **Error Analysis of SPGEM in the benchmark compositions-group-seen-colors.**

| Module | No Noise | Noise=0.01 | Noise=0.05 | Noise=0.1 | Noise=0.5 | Noise=0.7 |
|---|---|---|---|---|---|---|
| Visual Grounding | 77.4 | 77.1 | 77.3 | 77.0 | 75.8 | 69.4 |
| Goal Generation | 77.4 | 76.5 | 76.3 | 74.6 | 72.3 | 71.5 |

Table 10: **SPGEM's robustness to noise in the benchmark compositions-group-seen-colors.**

marized quantitatively in Table-10 and qualitatively in Figure-6. We observe that our model is quite robust to noise even as high as 0.5 the size of the object.

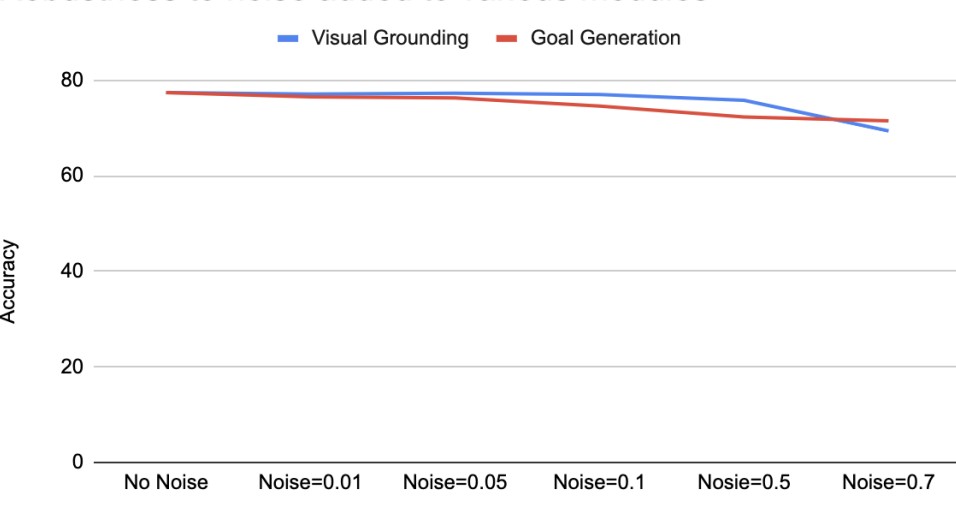

Figure 6: **Robustness to noise added to outputs from visual grounding and goal generation modules**

### 6.2.4 COMPARISON WITH HARDCODED GOALS

We experimented with adding hardcoded goal generation in place of our EBMs for our benchmarks. For binary relations and shapes we hardcoded the goal generation. However, for compositions, this is non-trivial because we would need to combinatorially search to find a goal configuration which would satisfy all relations. To sidestep this precise issue, while generating training demos in simulation, we randomly spawn some objects and subsample a bunch of relations from all possible $\binom{n}{2}$ binary relations to form the task description. Hence, for this hardcoded experiment, we directly take the final oracle goal from the generated data as the hardcoded goal. We call this a 'Combinatorial hardcoded goal'. Another reasonable hardcoding approach is to sequentially execute each hardcoded constraint. We call this 'Sequential hardcoded goal'. The quantitative results are in Table-11. We observe that on spatial relations, circle, lines and combinatorially hardcoded compositions, our model is only 3.6% worse on average than hardcoded goals. When compared on compositions with sequential execution, our model is 12.3% better since it generates a goal by considering all constraints jointly without doing any expensive combinatorial search.

| Benchmark | Hardcoded | Ours |
|---|---|---|
| Spatial Relations | 91.7 | 90.0 |
| Circles | 98.2 | 91.3 |
| Lines | 98.9 | 98.1 |
| Composition-one-step w/ combinatorial hardcoding | 95.1 | 91.0 |
| Composition-one-step w/ sequential hardcoding | 76.3 | 91.0 |
| Composition-group w/ combinatorial hardcoding | 76.9 | 72.4 |
| Composition-group w/ sequential hardcoding | 62.4 | 72.4 |

Table 11: **SPGEM's performance against oracle hardcoded goals.**

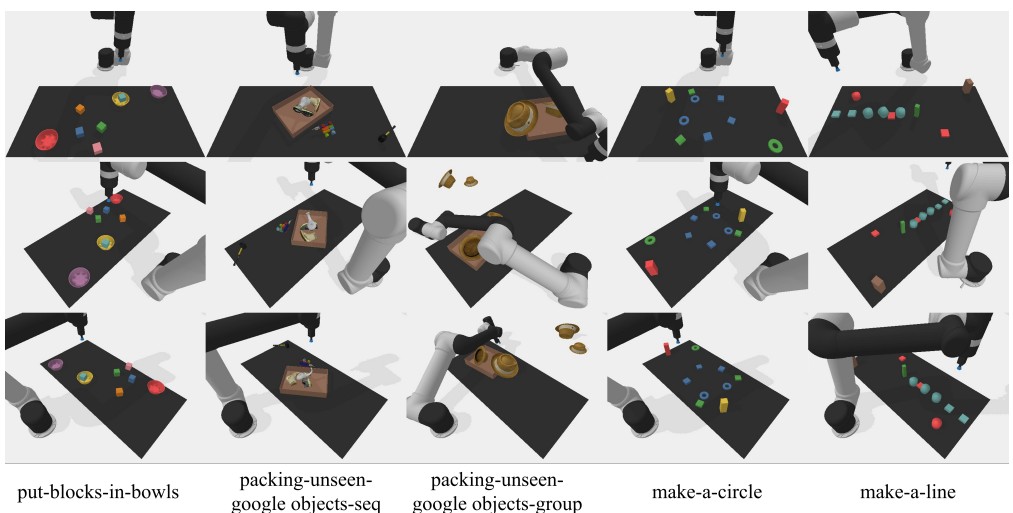

put-blocks-in-bowls  packing-unseen-google objects-seq  packing-unseen-google objects-group  make-a-circle  make-a-line

Figure 7: **Manipulation tasks in simulation.**

### 6.2.5 EVALUATION METRICS FOR REARRANGEMENT TASKS

For evaluating making-a-circle task, we fit a best-fit circle for the final configuration predicted by SPGEM. To do this, we consider the centers of the bounding boxes as points. Then, we compute the centroid of those points and the distance of each point from the centroid. This is an estimate of "radius". We compute the standard deviation of all this quantity. If this is lower than 0.03, then we assign a perfect reward. The reward linearly decreases when the std increases from 0.03 to 0.06. Beyond that, we give zero reward. We tuned these thresholds empirically by generating and distorting circle configuration.

We follow similar evaluation strategy for make-a-line. Here we compute the average slope and fit a line to our data. Then we measure the standard deviation of the distance of each point from the line. We found that the same thresholds we used for circles work well for lines as well.

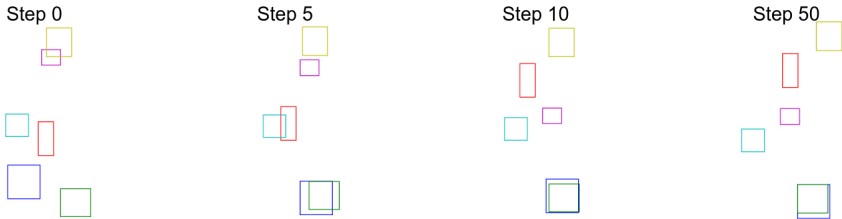

*"Place the blue box right of the red box and below the yellow box. Put the cyan box left of the blue box. Move the red box above the magenta box. Place the green box inside the blue box."*

Figure 8: **Spatial planning via composable concept graph energy minimization.**

### 6.3 LIMITATIONS AND PROMISING DIRECTIONS FOR FUTURE WORK

**Physics**: Our model do not take into account physics based priors while predicting a goal. Other related methods like CLIPort, Transporter or Language Planners also do not incorporate these priors and we believe incorporating physics into manipulation planning and policy learning remains an open research problem. Incorporating these priors are critical to identify if the generated configuration is stable or not. It would be even more useful if these priors can be added directly in the goal generation to directly avoid failures during execution. A promising direction is to model physics based constraints as energies. Naturally, one viable solution to achieving that is to leverage simulators that provide differentiable simulation, where we can additionally consider the gradients of the scene stability with respect to the spatial configuration of the scene when optimizing our EBMs. Recently, a number of such differentiable simulators have been proposed (Huang et al., 2021; Heiden et al., 2021; Qiao et al., 2020). Given such a simulator, we can jointly optimize for the physics based constraints and the constraints mentioned in the language jointly to achieve a physically stable goal configuration that our robots can execute.

**Temporal Ordering**: Our goal generation module outputs the final goal configuration for the given instruction, but does not tell the ordering of the subgoals to execute them. For example, when making a tower, the goal generator would place all objects into a tower but would not tell which objects to pick up first and thus inadvertently creating occlusions. One solution to this problem is to heuristically pick the order based on objects that are closer to the floor in the predicted scene configuration. However, a more general solution would be to add temporal constraints as energy priors which could be an interesting direction for future research.

**Location Parameterization**: Different tasks need different abstractions – while many pick and place tasks for rigid objects can be modelled by abstracting them away with bounding boxes, there are tasks that might need finer location parameterization like keypoints or object segmentation masks. The EBMs then sit on top of these finer parameterizations to optimize the energy for the given task. There are multiple ways of getting these parametrizations. The simplest way is to use off-the-shelf object detectors, object segmentors and keypoint detectors to get a hierarchy of parameterizations and then train a simple selection module which conditions on the task to identify the relevant parameterization. Another possible solution would be to design EBMs that sit on a hierarchy of parameterizations similar to the JEPA model proposed in (LeCun, 2022). We leave this exploration for future work.

### 6.4 EBM ENERGY LANDSCAPE VISUALIZATION

In this section, we visualize the energy landscape of our EBMs. For binary relations, we fix an object's bounding box in the scene and move the bounding box of the other object all over the scene and evaluate the energy of the configuration. We expect the energy to be low in regions which satisfy the relation w.r.t the fixed object and high in other regions. For shapes like circles and lines, we cannot directly plot the energy landscape because we would need to jointly consider all possible combinations of objects in the scene which would be impossible to represent in 2D or 3D landscape. Hence, we fix all but one object in a valid circle location and move the free box in all possible regions. We then expect the energy landscape to have low values in regions which complete the shapes and high otherwise. For compositions, we consider a single object's relation to all other objects and score the joint sum of energy for all constraints expecting the energy to be low in regions which satisfy all constraints and high otherwise.

The plots are shown in Figure-9. We observe that the energy landscape is usually smooth and has low energies in valid regions and high energy otherwise.

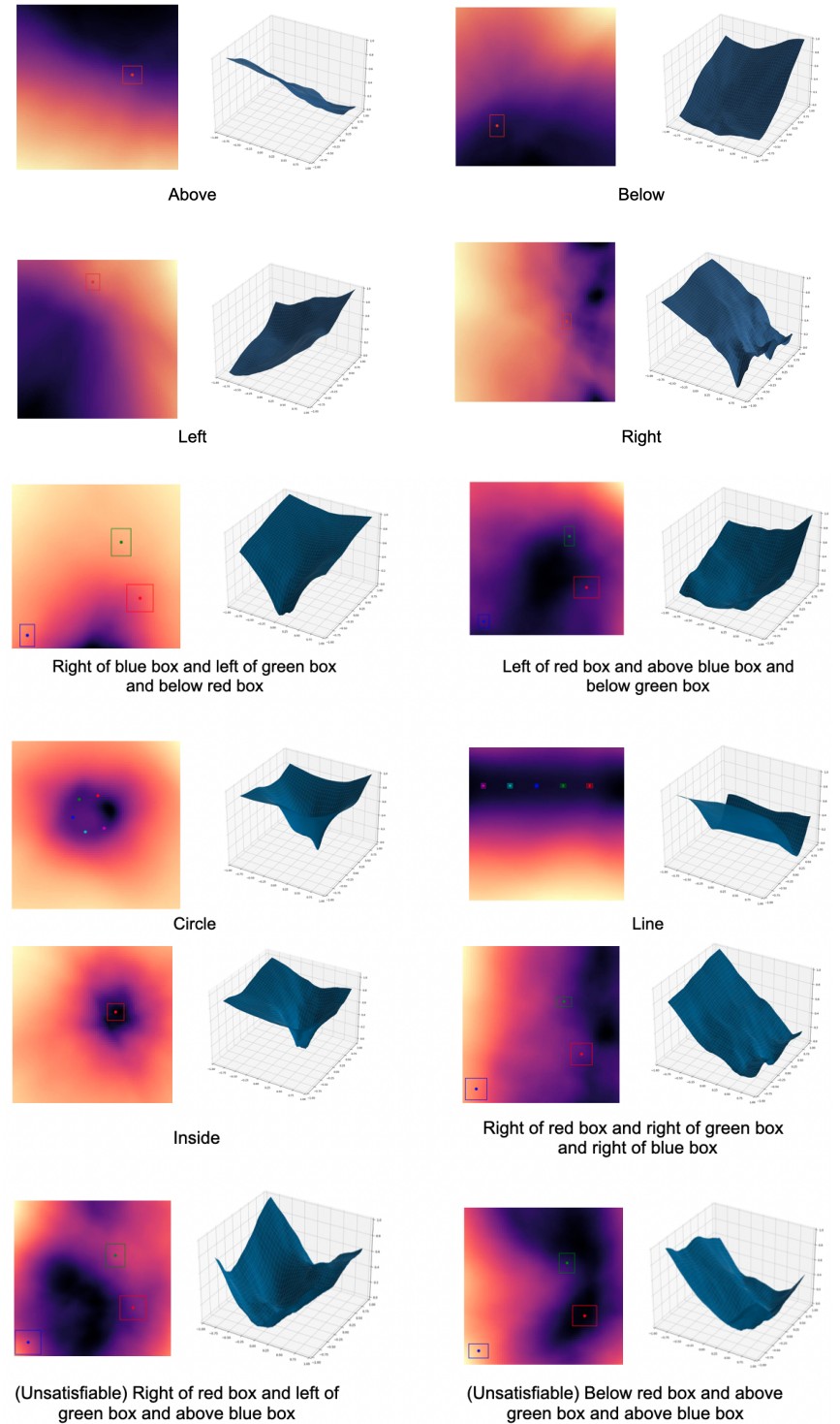

Figure 9: **EBM Energy Landscape visualization:** In this visualization, the boxes shown here remain fixed, and we score the energy of another box relative to the fixed box by moving it all along the workspace. Energy decreases from Red to black in this heatmap.

| Operation | Signature | Semantics |
|---|---|---|
| Filter | (ObjectSet, ObjectConcept) → ObjectSet | Filter out set of objects based on some *Object Concept* like object name (eg. cube) or property (color, material) |
| BinaryEBM | (Object A, Object B, Relation) → (Pick locations, Place locations) | Executes *BinaryEBMs* for rearranging Object A and Object B to satisfy the given binary *relation* (like left of/right of/inside etc.) |
| MultiAryEBM | (ObjectSet, Shape Type, Property) → (Pick locations, Place locations) | Executes *MultiAryEBMs* for the given *Shape Type* (circle, line, etc.) with specified *Properties* (like size, position etc.) on a set of given objects and generates pick and place locations to complete the shape. |

Table 12: All operations in the domain-specific language for SPGEM

## 6.5 BENCHMARK GENERATION

We extend the Ravens (Zeng et al., 2020) and CLIPort benchmark (Shridhar et al., 2021) for spatial reasoning in the PyBullet simulator. For each benchmark, we write a template sentence (eg: "Arrange OBJ1 into a circle") and then randomly select valid objects (and colors) from a pre-defined list. To test generalization, we include novel colors or novel objects in the evaluation set. Once the sentence is generated, we programatically define valid regions which satisfy the relation and then sample empty locations from it to specify object goal locations. We start by placing all objects randomly in the scene and then an oracle, hand-designed, policy picks and places the objects to the desired locations and returns a demo trajectory which involves the raw RGB-D images and pick and place locations. These can be used then to train a behaviour cloning policy similar to CLIPort.

## 6.6 ETHICAL STATEMENT

Mapping language instructions to goals for robot coaching would contribute to the development of intelligent agents that are programmable using natural language, as opposed to being programmed explicitly or through positive and negative examples. This will facilitate human-robot interaction. Robots will then be able to acquire new concepts and commands on-the-fly, adapting to every user's needs and desires, as opposed to maintaining a fixed set of functionalities.

With the potential benefits of easier communication with artificial agents, however, come some drawbacks. An increased ability to program robots would also be followed by more automation and less demand for employees, even for jobs that are now considered "safe". While in the long term, automation would probably be beneficial to society by increasing productivity, in the short term, the process of automation could have a negative impact on many people.

Lastly, while we aim to ease the programming of a robot, we cannot have control over the purposes this technology will be used, e.g. whether a robot is tasked for beneficial or catastrophic purposes. Our aim is to enable the technology's potential for social good, such the potential of non-expert users, e.g. elderly people, to instruct a robotic assistant.

