# OpenReview forum: "Spatial reasoning as Object Graph Energy Minimization"
_ICLR.cc/2023/Conference — Submitted to ICLR 2023_

### Official Review · Reviewer_erLa · 2022-10-25

**Confidence:** 3
**Correctness:** 3
**Technical Novelty And Significance:** 2
**Empirical Novelty And Significance:** 2
**Recommendation:** 5

**Clarity, Quality, Novelty And Reproducibility:**

### Clarity
More detail should be provided for the core contribution of the paper (the EBM-based optimization scheme).

### Quality
OK. See weaknesses above for my major concerns.

### Novelty
Optimization-based scene rearrangement is not new, there has been decades of research on this topic (see weaknesses above). Learnable EBM seems novel to me, at least with respect to the specific task, but whether they are useful is questionable.

### Reproducibility
Code is provided, looks reasonable, but probably need a bit of cleanup and a README file.

**Strength And Weaknesses:**

#Strength
- Good empirical performance on the set of tasks evaluated. Although these tasks are rather biased towards the proposed method, it is still nice to see.
- Optimization guided by learnable priors is a good idea and should be fairly generalizable. The ability to learn these priors with a simple MLP over object attributes is also nice to know.

#Weaknesses
### Novelty
- Constraint-guided layout optimization is a long studied problem, see earlier works such as "Make it home: automatic optimization of furniture arrangement", "Interactive Furniture Layout Using Interior Design Guidelines" for optimization, and works such as "Example-based Synthesis of 3D Object Arrangements" for learnable spatial priors. In some sense, the types of constraints dealt with in these works are much more complex than those in this work.
- While learnable spatial priors are nice, the set of relations dealt with these work is rather simple, so it is unclear whether the techniques introduced here can apply to more complex scenarios (many work suggests that neural networks struggle to perform complex spatial reasoning, especially if based on axis-aligned bounding boxes).
- It is not clear to me what is the significance of the language to constraint module and the robotic planning module. Seems that the work can exist without either of these and neither adds much to the contribution of this work.

### Design Choices
- I am not sure whether the proposed EBMs can generalize to more complex spatial relations, especially if semantics are involved. Relations such as "to the left of" are rathe easy to learn and I am not seeing examples of more complex cases e.g. circle. It seems that the benefit of a learnable prior really starts to emerge for more complex cases, whereas the type of constraints studied in this paper can be easily hardcoded (in a way that allows a gradient-descent based optimization scheme).
- I am concerned about the optimization landscape formed by the union of all EBMs. It seems that there can easily be scenarios where the optimization get stuck in a local minimum (e.g. object A starts above B, needs to move below B, but additional constraints enforce that A cannot appear to either the left or the right of B). There is a reason why many of these works I mentioned above resorts to methods more complex than simple hill climbing e.g. MCMC with large jumps, simulated annealing, etc.
- It seems that the method will struggle in cases where not all constraints can be satisfied. Since the learned functions lacks a consistent magnitude, it can be tricky to balance the weights of different constraint and ensure that the most number of constraints can be satisfied.

### Evaluation
- Would be really helpful to visualize the learned energy function. Since the constraints studied here is rather simple, the models really don't need to learn much to lead to good results. A visualization showing a smooth function that consistently satisfies the constraints can increase my confidence in the proposed method.
- A baseline with hardcoded rules is needed - the method won't be of much use if it performs much worse than such a baseline.

**Summary Of The Paper:**

This paper proposes an optimization scheme to arrange a set of objects so they conform to a set of spatial constraints, that are either binary directional relationships, or specific spatial arrangements e.g. line or circle. An energy based model (EBM) is learned for each of the constraint type, which maps a set of obvious bounding boxes and locations to score that indicates how well the constraint is satisfied. These models are learned by enforcing that the ground truth layouts scores higher than perturbed layouts. At test time, a gradient-descent scheme is used to optimize the position of the objects, with the guide of the learned EBMs. The paper also incorporates off-the-shelf methods that maps language to constraints, and uses the optimized layout to guide a robotic agent. However, I am not sure if either of these relates to the central contribution of the paper. Evaluation show that the proposed method performs better than general purpose embodied agents on a set of tasks that seem to be tailored towards the proposed method.

**Summary Of The Review:**

I don't know much about robotics, but it appears to me that the core of the method is the learned-prior-guided layout optimization. This works without visual language grounding or the transporter network. I have many concerns with this optimization scheme, especially if it can generalize to more complex scenarios (see weaknesses above). As a result, while I think this method can be potentially useful, I lean towards rejection.

---

> ### Author Response · Authors · 2022-11-16
> **Response to Reviewer erLa (Part 1)**
>
> Thank you for your comments. We address your concerns below:
>
> > It is not clear to me what is the significance of the language to constraint module and the robotic planning module. Seems that the work can exist without either of these and neither adds much to the contribution of this work.
>
> We understand your concerns regarding the connection between the language to constraint module and the robotic planning setting. We would like to re-emphasize that the main problem we are trying to address in this work is following language instructions for spatially rearranging scenes in the context of robotic manipulation. Improving spatial reasoning for robot manipulation is a central motivation of our paper. Existing methods either learn to map instructions to object goal locations and actions directly, or prompt pre-trained Large Language Models (LLMs) to decompose instructions into sequences of more straightforward language goals to be fed to reactive language-to-action policies. The former have shown limited generalization to environment and instruction variations, and the latter suffer from language bottleneck during the planning and describing subgoals in language space -- for example, it is difficult to explain in language a "circle" or subgoals for the composition of several spatial concepts. **Hence, our main insight is to plan in the visual space for robot manipulation tasks.** As a means to this end, we propose to use a library of Energy-based models (EBMs) and show a complete system involving grounders, parsers, EBMs and a low-level policy for language-guided robot manipulation. We show our model can execute highly compositional instructions zero-shot in simulation and in the real world and outperforms state-of-the-art language-to-action models and LLM-based planners by a margin in simulation, especially for long instructions and multi-ary spatial concepts. Thus, while energy optimization is a central component of our method, **showing its applicability in robot manipulation** is an equally important aspect of our work.
>
> ***
>
> > Constraint-guided layout optimization is a long studied problem, see earlier works such as "Make it home: automatic optimization of furniture arrangement", "Interactive Furniture Layout Using Interior Design Guidelines" for optimization, and works such as "Example-based Synthesis of 3D Object Arrangements" for learnable spatial priors. In some sense, the types of constraints dealt with in these works are much more complex than those in this work.
>
> Thank you for pointing us to these works, we have added them to our related works. Here is a draft of the paragraph we added:
>
> "Automatic optimization for object re-arrangement has been studied outside the field of robotics. [1] and [2] use a few user-annotated examples of scenes to adapt the hyperparameters of task-specific cost functions, which are then minimized using standard optimization algorithms (hill climbing and/or simulated annealing). To learn those hyperparameters from data, these approaches fit statistical models, e.g. mixtures of Gaussians, to the given samples. [3] further employ such optimization constraints into an interactive environment, where the user can provide an initial layout and the algorithm suggests improvements. All these approaches require expert knowledge to manually design rules and cost functions, namely [1] identifies seven and [3] eleven expert-suggested criteria for successful re-arrangement. Since they are hand-crafted, these methods do not generalize beyond the domain of furniture arrangement. In contrast, the energy optimization we propose is purely data-driven and domain-agnostic: a neural network scores layouts, assigning high energy to those that do not satisfy the (implicit) constraints and low energy to those who do, essentially modelling the underlying distribution of valid layouts."
>
> While we do agree that constraint-guided layout optimization and scene generation are well-studied problems, **our goal is to make scene generation useful for spatial reasoning and instruction following**. Also, although far from the scope of this work, we believe that an energy-based model that absorbs semantic and pose information of an object configuration can learn to generate plausible layouts for furniture as well. We show some preliminary **experiments with EBMs altering object poses** later in this discussion.
>
> [1]: Yu, Lap Fai, et al. "Make it home: automatic optimization of furniture arrangement." ACM Transactions on Graphics (TOG)-Proceedings of ACM SIGGRAPH 2011, v. 30,(4), July 2011, article no. 86 30.4 (2011).
>
> [2]: Merrell, Paul, et al. "Interactive furniture layout using interior design guidelines." ACM transactions on graphics (TOG) 30.4 (2011): 1-10.
>
> [3]: Fisher, Matthew, et al. "Example-based synthesis of 3D object arrangements." ACM Transactions on Graphics (TOG) 31.6 (2012): 1-11.
>
> ***

---

> > ### Author Response · Authors · 2022-11-16
> > **Response to Reviewer erLa (Part 2)**
> >
> > > While learnable spatial priors are nice, the set of relations dealt with these work is rather simple, so it is unclear whether the techniques introduced here can apply to more complex scenarios (many work suggests that neural networks struggle to perform complex spatial reasoning, especially if based on axis-aligned bounding boxes).I am not sure whether the proposed EBMs can generalize to more complex spatial relations, especially if semantics are involved. Relations such as "to the left of" are rather easy to learn and I am not seeing examples of more complex cases e.g. circle. It seems that the benefit of a learnable prior really starts to emerge for more complex cases, whereas the type of constraints studied in this paper can be easily hardcoded (in a way that allows a gradient-descent based optimization scheme).
> >
> > - **Complex Scenarios**:
> >   - **"eg. circle"** : In the main paper, **we have a benchmark of spatial-relations which involves making circles and lines.** For quantitative results on them, please refer to Table 3. For qualitative results, we invite you to check our [website](https://sites.google.com/view/spgem/) which shows qualitative examples of making circles and lines in both simulation and in a real-robot setup. In each video, we visualize all intermediate steps involving grounding and energy optimization. We also **show (zero-shot) results for the "circle inside a plate" involving the composition of the circle and inside concept.** Please also check the ["EBM Library Section"](https://sites.google.com/view/spgem/#h.2gkbyichrj6h) and ["EBM Energy Landscape Visualization"](https://sites.google.com/view/spgem/#h.sr46fpckkhmc) sections on our website for more visualizations.
> >   - **"type of constraints studied in this paper can be easily hardcoded"**: While it is true that we can hardcode simple relations like "left", "right", "above", "and below"; hardcoding compositional relations is harder because that would need a combinatorial search over all constraints to find a feasible solution set. In contrast, via EBMs we can simply perform gradient descent over the sum of energies for all constraints. Additionally, our goal is learning robot manipulation from  demonstrations. Most humans would not be able to code in order to teach their robots new concepts while creating demos as we used in our setup is easy for anyone. Thus, hardcoding the constraints would defeat that purpose.
> >   - **"many work suggests that neural networks struggle to perform complex spatial reasoning, especially if based on axis-aligned bounding boxes"**: As we also note in our limitations and discussed further in Section 6.3 of our Appendix, we do agree that axis-aligned bounding box as an object abstraction would not suffice for many tasks, and extending this work to consider more complex parameterization is a direct avenue for future work. However this is not a fundamental limitation of EBMs, they can be extended to consider abstractions beyond axis-aligned boxes. As a proof of concept, **we trained EBMs to optimize the orientation of posed objects with respect to reference points**, a setup which is common beyond robotics as well, e.g. “the armchair should look towards the tv”. Please check our website, Section [“EBMs with Pose Information”](https://sites.google.com/view/spgem/#h.8rqgcnis3ddy) for interesting qualitative results of these EBMs. We additionally refer to Section 6.1.2 in the appendix for implementation details.
> > ***
> >
> > > I am concerned about the optimization landscape formed by the union of all EBMs. It seems that there can easily be scenarios where the optimization get stuck in a local minimum (e.g. object A starts above B, needs to move below B, but additional constraints enforce that A cannot appear to either the left or the right of B). There is a reason why many of these works I mentioned above resorts to methods more complex than simple hill climbing e.g. MCMC with large jumps, simulated annealing, etc
> >
> > Indeed, the optimization landscape can have local minima, even in the case of single relations, e.g. see the landscape for “right” in Section [“EBM Energy Landscape Visualization”](https://sites.google.com/view/spgem/#h.2gkbyichrj6h) of our website. This reflects the biases of the small set of demonstrations that we used. To address this fact, **our optimization process is not deterministic,** as we add random noise at every timestep. The amount of noise is higher in the first steps of the optimization to promote exploration, then it is gradually reduced as in Simulated Annealing. Equation 1 in the main paper reflects that. We updated Section-3 to refer to that connection.
> >
> > ***

---

> > > ### Author Response · Authors · 2022-11-16
> > > **Response to Reviewer erLa (Part 3)**
> > >
> > >
> > > >It seems that the method will struggle in cases where not all constraints can be satisfied. Since the learned functions lacks a consistent magnitude, it can be tricky to balance the weights of different constraint and ensure that the most number of constraints can be satisfied.
> > >
> > > Our energy values are not explicitly normalized to a fixed scale, so what you are describing is a valid concern. In practice, **we find that for our five binary concepts the range of the energy values is similar, from -0.02 to 0.15**. This is likely because of the L2 regularization loss [1] we add over the energy values during training which keeps the energy of each EBM near zero. As a result, we find that **directly summing the energies without any extra weighing works well** even in cases where not all constraints can be satisfied. We showcase this in [Section "EBM Energy Landscape Visualization"](https://sites.google.com/view/spgem/#h.sr46fpckkhmc) of our website in the bottom row where we visualize the energy landscape for two descriptions where only two out of the three relations can be satisfied. We observe that the energy landscape remains smooth but with multiple local minima -- which indeed satisfy the most number of constraints.
> > >
> > > [1] Du et al. 2021, Improved contrastive divergence training of energy based models
> > >
> > > ***
> > >
> > > > Would be really helpful to visualize the learned energy function. Since the constraints studied here is rather simple, the models really don't need to learn much to lead to good results. A visualization showing a smooth function that consistently satisfies the constraints can increase my confidence in the proposed method.
> > >
> > > Thank you for the suggestion!
> > > - We **added energy landscape visualization as a 2D heatmap and 3D Energy plots** on [our website in Section "EBM Energy Landscape Visualization"](https://sites.google.com/view/spgem/home#h.sr46fpckkhmc) and Figure-10 of our Appendix. We visualize them for all binary spatial relations (like left, right, above, and below); for shapes like "circles" and "lines"; and for the composition of spatial relations. For more details, please refer to Section 6.4 in our Appendix.
> > > - Besides these, we **added EBM energy optimizations** in all our [simulated](https://sites.google.com/view/spgem/home#h.lopaf7ccl6xp) and [real robot](https://sites.google.com/view/spgem/home#h.g0v2pevzi5gc) qualitative results which are also [available on our website.](https://sites.google.com/view/spgem/)
> > >
> > > Please feel free to suggest any other visualizations that you would like to see!
> > >
> > > ***
> > >
> > > > A baseline with hardcoded rules is needed - the method won't be of much use if it performs much worse than such a baseline.
> > >
> > > Thank you for the suggestion! As discussed earlier, while our goal is learning spatial concepts from demonstrations, we agree that comparison with a hardcoded baseline is important. Towards that end, **we experimented with adding hardcoded goal generation in place of our EBMs for our benchmarks**. For binary relations and shapes, we hardcoded the goal generation. However, for compositions, this is non-trivial because we would need to combinatorially search to find a goal configuration which would satisfy all relations. To sidestep this precise issue, while generating training demos in simulation, we randomly spawn some objects and subsample a bunch of relations from all possible <sup>n</sup>C<sub>2</sub> binary relations to form the task description. Hence, for this hardcoded experiment, we directly take the final oracle goal from the generated data as the hardcoded goal. We call this a `Combinatorial hardcoded goal`. Another reasonable hardcoding approach is to sequentially execute each hardcoded constraint. We call this a `Sequential hardcoded goal`. The quantitative results are in Table 11 of our Appendix. We observe that on isolated single spatial relations, circles, lines and combinatorially hardcoded compositions, **our model is only 3.6% worse on average than hardcoded goals.** When compared on compositions with sequential execution, **our model is 12.3% better** since it generates a goal by considering all constraints jointly without doing any expensive combinatorial search. We also attach a [barplot of this comparison on our website](https://sites.google.com/view/spgem/home#h.pm6r6ow15x47) for easy exposition.
> > >
> > > We want to re-emphasize that hard coding for compositions is not at all trivial and scales exponentially with the number of constraints. Besides, as we explained earlier, hardcoding goal generation severely impacts the model's capability to learn directly from (human) demonstrations.
> > >
> > > ***
> > >
> > > >More detail should be provided for the core contribution of the paper (the EBM-based optimization scheme).
> > >
> > > Per your suggestion **we added more details on the EBMs** in the main paper and (mainly) in Section 6.1.2 of the appendix.
> > >
> > > ***

---

> > > > ### Author Response · Authors · 2022-11-16
> > > > **Response to Reviewer erLa (Part 4)**
> > > >
> > > > >Code is provided, looks reasonable, but probably need a bit of cleanup and a README file.
> > > >
> > > > We plan to release our code, well-documented and clean, on GitHub including our data and checkpoints, upon publication. Meanwhile, per your suggestion, we cleaned it up a bit more and added instructions on running the experiments in README.
> > > >
> > > > ***
> > > >
> > > > Thank you for your valuable suggestions and we are looking forward to your further feedback!

---

> > > > > ### Author Response · Authors · 2022-12-03
> > > > > **A gentle reminder for post-rebuttal discussion**
> > > > >
> > > > > Dear reviewer erLa,
> > > > >
> > > > > Thank you for your effort in reviewing our paper. Following our discussion, we revised our paper accordingly to
> > > > > - include a hard-coded baseline
> > > > > - visualize the optimization process in both simulation and real-world experiments
> > > > > - showcase the generality of EBMs by extending them to optimize the pose of objects
> > > > >
> > > > > Since few days are remaining for the discussion phase, we would appreciate it if you let us know whether you have any further questions.
> > > > >
> > > > > Sincerely,
> > > > >
> > > > > anonymous authors

---

> > > > > > ### Comment · Reviewer_erLa · 2022-12-09
> > > > > > **Response**
> > > > > >
> > > > > > Sorry for the delayed response. I did read over the response a while ago and was overall satisfied with it. Didn't really have time to write my thoughts down, but I don't really have any additional follow up questions anyways. Will update my score after the reviewer meeting.
> > > > > >
> > > > > > Re: responses:
> > > > > >
> > > > > > (language to constraint and robotic planning): I agree that there is value in showing the applicability of the proposed method in these scenarios. However, If I were to write this, I would still focus less on these aspects: any layout optimization algorithm can be applied to these scenarios, overly focus on these does distract the readers away from the core contribution of this paper --- my initial impression of this paper was lower precisely because of such distractions.
> > > > > >
> > > > > > (prior work in layout generation / scene synthesis): i'm fine with not discussing them in related works. That's a pretty long line of work anyways and recent works are indeed much more data driven, though they often rely less on optimization... I respectfully disagree with the claim about these method not being able to generalize: the arrangement model of [3], for example doesn't really assume much domain specific knowledge. My point about bringing up these works is more about providing a handle towards a line of work that's relevant to this paper, and hopefully you can find some interesting ideas in these three works and their follow up.
> > > > > >
> > > > > > (Complex scenarios): I checked the example and they do look decent. However, these cases (Manhattan lines, circles) are still relatively straightforward to define. I don't think you really have the burden to showcase more complex scenarios, but I think it would be interesting to explore the possibility to use the model to handle more complex cases e.g. something like a line of a particular slope, a ellipse, or extending to 3D. Would also be interesting to see if, given appropriate labeled data e.g. COCO/CLEVR/3D scene datasets, it can learn to encode more semantic relationship as well (I think this is the part where the language to constraint component can become really interesting). Still, I think these should be regarded as future works - the additional examples shown in the rebuttal definitely made me more confident in this work - after all, neural implicits have become the dominating paradigm in modeling 3D geometry, and it is very reasonable that something similar can work better as well for modeling spatial relations.
> > > > > >
> > > > > > (Optimization landscape / weighting / visualization of learned rules) The visualizations are great and did convince me. Again, seeing them for more complex cases would have been better, but it's unreasonable to ask for everything for a paper that takes a first step.
> > > > > >
> > > > > > (Additional baseline) This looks good. I think there are some ways to use hard coded rules for compositional cases as well, but I think the baselines are good enough.

---

### Official Review · Reviewer_MYGL · 2022-10-29

**Confidence:** 3
**Correctness:** 4
**Technical Novelty And Significance:** 3
**Empirical Novelty And Significance:** 3
**Recommendation:** 5

**Clarity, Quality, Novelty And Reproducibility:**

The paper is clearly written, and the concept of object graph optimization is useful under the setting of language to action. The work is reproducible. A more focused contribution, in depth analysis of the core idea, and more sophisticated experimental settings would make the paper more interesting to read.

**Strength And Weaknesses:**

Strength
=======
- A nice formulation of the spatial arrangement as object graph energy minimization. Although this treatment has been studied for decades, it is still interesting in the context of language instruction for robotics. The setting also requires a minimal amount of training data, as most heavy lifting parts (linguistic, visual and robotic) are handled by existing systems.
- The empirical results are promising.

Weaknesses
==========
- The limitations discussed in the paper (temporal constraints, location parameterization, simple predicates) are important to address, even though energy minimization can be theoretically suitable for solving these problems.
- There are many moving parts in the overall system, it is important to see how robust the system is if the parts are not too reliable (e.g., semantic parsing, object grounding, robotic failure).

**Summary Of The Paper:**

The paper formulates the problem of spatial arrangement of objects as minimizing the energy function over the configurations of the object graph. The graph is constructed by parsing the language instruction and utilizing visual language grounding. The graph energy function is composed of predicates parameterized as neural networks. The model is trained using contrastive divergence with Langevin dynamics. Finally real object arrangement is done by a Transporter Network. Empirical results show that this formulation exhibits zero-shot generalization capability and works better than existing methods on the tasks evaluated.

**Summary Of The Review:**

The paper introduces a way to treat of spatial arrangement as object graph energy minimization under the language to action context. The formulation itself isn't new, but the overall solution is interesting and applicable in certain settings. The are important limitations, however.

---

> ### Author Response · Authors · 2022-11-16
> **Response to Reviewer MYGL**
>
>
> >There are many moving parts in the overall system, it is important to see how robust the system is if the parts are not too reliable (e.g., semantic parsing, object grounding, robotic failure).
>
> Per your suggestion, we conducted a **detailed error analysis for our model** identifying the errors contributed by each of the modules i.e. semantic parsing, grounding, goal generation and robotic failure on composition-group-seen-colors benchmark in Table 9 of the Appendix. We observe that 6.0% of the errors are due to failures in robot execution, 11.7% due to goal generator failure, 5.1% due to grounding errors while language parsing do not contribute to any errors.
>
> Each of the four components of our model is critical for the task's success. These modules are robust to minor errors like a) the grounding model predicting a noisy box (for example smaller or larger than the actual object size) and b) EBM predicting a goal which is slightly off. For example, if the task is to place a cube in a bowl, and EBM generated a goal location which partially includes the area outside the bowl, the low-level policy is trained to predict a location inside the bowl making it robust to such perturbations. Inspired by your comment, we did **systematic robustness analysis** by adding Gaussian noise of varying magnitudes to each of the grounding and goal generation modules predictions. The quantitative results can be found in Table 10 in Appendix and are also visualized on our website in Section ["Robustness to noisy outputs"](https://sites.google.com/view/spgem/home#h.ce3n9fu7zgp4). We find that **our model is robust to even high magnitudes of added noise.**
>
> We add these experiments and analysis in Section 6.2.3 of our Appendix.
>
>
> If any of the modules completely fails i.e. if parsing is incorrect, or if the grounder misdetects or if the EBM predicts the wrong goal or if there is a robot failure, then the final task would also likely fail. We invite you to check the goal detection feedback and recovery modules that we are adding to the paper in Section 3, a detailed discussion on feedback in Section 6.2.2 in Appendix and [some visualizations on our website](https://sites.google.com/view/spgem/home#h.bsv3bgk7x1mz) (also discussed further in response to Reviewer hrSa). These modules would help in making our model more robust.
>
> ***
>
> > The limitations discussed in the paper (temporal constraints, location parameterization, simple predicates) are important to address, even though energy minimization can be theoretically suitable for solving these problems.
>
> Thank you for the suggestion. We **added a detailed section discussing the limitations and promising line of attacks** for a) Physics and b) Location parameterization and c) Temporal Ordering in Appendix section 6.3. Specifically for the location parameterization limitation, we also **added qualitative results with EBMs which optimize over posed boxes** on our [website](https://sites.google.com/view/spgem/home#h.8rqgcnis3ddy) and discuss them in Section 6.1.2 of the appendix.
>
>
> ***
>
> > A more focused contribution, in depth analysis of the core idea, and more sophisticated experimental settings would make the paper more interesting to read.
>
> Thank you for suggesting this. For more in-depth analysis, besides the aforementioned error analysis and robustness experiments, we added:
> - a **detailed generalization test benchmark** with [a graph of the performance comparison here](https://sites.google.com/view/spgem/home#h.ij4y30ojzm2x) in addition to detailed results in Table 6 and analysis in section 6.2.1 of our Appendix.
> - Per the suggestion of reviewer erLa, we also **added comparisons with hardcoded goal baselines** in Section 6.2.4 and [on our website.](https://sites.google.com/view/spgem/#h.pm6r6ow15x47)
> - **Detailed visualizations of Energy Landscapes** on our [website](https://sites.google.com/view/spgem/home#h.sr46fpckkhmc) and in Section 6.4 of our Appendix.
>
> To address your comment about more sophisticated experimental settings:
> - We **added real-robot experiments**. Please find qualitative videos for compositions and shapes benchmark on our [website](https://sites.google.com/view/spgem/home#h.g0v2pevzi5gc). We show robot execution on complex instructions involving jointly satisfying multiple spatial relations. We also attach a video for “Rearrange all small objects in a circle inside the plate” showing zero-shot composition of n-ary (circle) and binary concepts (inside).
> - We also **added qualitative results with EBMs which optimize over posed boxes** on our [website](https://sites.google.com/view/spgem/home#h.8rqgcnis3ddy).
>
> ***
>
> Thank you again for all your useful feedback! We would be happy to answer more questions or incorporate any other feedback in our paper.

---

> > ### Author Response · Authors · 2022-12-03
> > **A gentle reminder for post-rebuttal discussion**
> >
> > Dear reviewer MYGL,
> >
> > Thank you for your effort in reviewing our paper. Following our discussion, we revised our paper accordingly to include
> > - a detailed error and robustness analysis for our model
> > - real-world experiments
> > - more discussion and insights on our limitations, including an illustration of the generality of EBMs by extending them to optimize the pose of objects
> >
> > Since few days are remaining for the discussion phase, we would appreciate it if you let us know whether you have any further questions.
> >
> > Sincerely,
> >
> > anonymous authors

---

### Official Review · Reviewer_hrSa · 2022-11-01

**Confidence:** 4
**Correctness:** 3
**Technical Novelty And Significance:** 3
**Empirical Novelty And Significance:** 3
**Recommendation:** 6

**Clarity, Quality, Novelty And Reproducibility:**

Clarity: The paper reads well, barring some minor comments, detailed below.

Quality & Novelty: I have discussed the strengths and weaknesses above. I think the basic idea is definitely novel. But, it may not be easy to extend to real-world scenarios.

Reproducibility: Authors have shared the code in supplementary, and is willing to share the code upon publication.

Minor Comments:
1. Page 5: "For detailed …" It is slightly unclear to me, whether there is any other complexities involved while minimizing the sum of energy functions. It is better to explain at least in Appendix rather than sending the readers directly to the code.

2. Page 5: "Referential …" Parts of the implementation and method description is sometimes intertwined. Might be helpful to summarize the whole process as an algorithm, provided the modular nature of things.

3. Page 6: "scene rearrangement …" Its a bit out of the blue. How do you create this benchmark? You can describe it in short at the least.

4. In Table 1, what is meant by "Task progress"?


**Strength And Weaknesses:**

The idea about moving from spatial reasoning in language space to space of object 2D 3D coordinates and representing predicates though EBMs are quite interesting. I feel there are some challenges to scale the work in real-world settings. Also, a few aspects of the generalization claims seem unsupported through experiments.

A few concerns:
1. I am wondering is the set of binary and n-ary spatial concepts open? You at least need the knowledge of which is a binary vs n-ary predicate? How do you infer that in care you are not using a list of predicates? I see you are training one EBM per predicate, which means the predicate list should be fixed right?

2. In general, works such as Palm-Saycan [1] have also showed at least one experiment on real environments, such as kitchen environment, moving from table-top ones. As far as I remember, even LLMPlanner [2] uses a real-world table-top environment. I think, showing the work in such setting would help; especially given questions such as the first one.

3. Generalization claim: Also, I do not see experiments performance over novel objects are shown. Among novel attributes, do you only mean shape and color? Is there anything else considered?

4. LLMPlanner such as PaLM-saycan uses success predictors to give some feedback whether goal state has been reached.
Though such goal state detection through energy minimizer is discussed, I do not see whether such a feedback mechanism exist? Some part has been discussed as physics based issues in Limitations. However, this is important, as the model only outputs final goal configuration; implying many goal configurations may be hard to achieve through pick-and-place policies in reality. What is the effect of this?

[1] Do As I Can, Not As I Say: Grounding Language in Robotic Affordances, Ahn et al. 2022

[2] Inner Monologue: Embodied Reasoning through Planning with Language Models

**Summary Of The Paper:**

Authors use energy based models (EBMs) and object graph minimization to perform instruction-guided spatial rearrangement. Authors use EBMs to represent each spatial predicate (binary or n-ary). Objects are represented using the 3D or 2D overhead box coordinates. Authors train the EBMs using gradient descent on the sum of energies with respect to the coordinate space, providing the final configurations that best satisfies the instructions.
Using this, essentially, the model reason/plan over a pixel-abstract, but bounding-box aware space to interpret the instruction, predicting a goal state. Using a low-level and pick-and-place policy execution, the objects are finally shifted to the goal position.

Authors make following empirical claims:
1. The model outperforms the baselines, especially in complicated instructions
2. Generalization over  1) novel compositions,  2) novel instructions, and 2) novel objects and attributes.

**Summary Of The Review:**

The idea of representing spatial predicates as EBMs is novel; alongwith reasoning over the 2D or 3D object coordinate space for instruction-guided spatial rearrangement. I also agree that language space may not be the best option to reason spatially about images. However, some experiments are lacking to support the generalization claims about generalizing to novel objects and attributes, alongwith experiments in real-world setting. Training each EBM to correspond to a 2-ary or n-ary predicate also raises the question of how many spatial relations that this model can support. Also, given the goal configuration is output and needs to be executed by a low-level policy executor, its easy to think of scenarios where the executor may not be able solve this using a greedy policy (stacks, tower of hanoi, inadvertently creating obstacles etc.). So, I am borderline on this. I feel some more experimentation is required, some of which may be non-trivial.

---

> ### Author Response · Authors · 2022-11-16
> **Response to Reviewer hrSa (Part 1)**
>
> Thank you for your detailed and encouraging feedback on our work. We address your concerns below:
>
>
> > In general, works such as Palm-Saycan [1] have also showed at least one experiment on real environments, such as kitchen environment, moving from table-top ones. As far as I remember, even LLMPlanner [2] uses a real-world table-top environment. I think, showing the work in such setting would help; especially given questions such as the first one.
>
>
> Per your request, **we added an evaluation of the trained model on real-world experiments** of instruction-guided tabletop manipulation. Our real-world test set includes 10 instances of instruction-guided manipulation tasks in each of the three settings---defined in the main paper---a) Composition-one-step b) Composition-group and c) Shapes like circles and lines. We show the quantitative results of this evaluation in Table 5 of the main paper. The proposed model achieves a **high success rate (average 86.4% over four tasks) in the real world** without any finetuning. We believe this is due to the modular nature of the model; our open vocabulary detector is trained in abundance of passive data, the goal-predicted EBMs operate over object abstractions, not appearances, and the low-level pick-and-place policy modules need only to solve a simple local selection of pick and place locations within an image patch cropped out of the predicted pick and place boxes.
>
> We also show videos of robot execution of complex instructions in our [website](https://sites.google.com/view/spgem/home#h.g0v2pevzi5gc).
>
> ***
>
> >Generalization claim: Also, I do not see experiments performance over novel objects are shown. Among novel attributes, do you only mean shape and color? Is there anything else considered?
>
> In the paper, we only show results on novel object colors.
> Per your suggestion, we added a detailed analysis of generalization in Section 6.2.1 of our Appendix. In summary, we added:
>
>
> - **An exhaustive generalization benchmark**: We study the generalization of our model and the CLIPort baseline across three axes: a) Novel object colors, b) novel table's background color, and c) novel object instances, in the spatial relations and composition benchmarks. In each of these settings, we only change one attribute (object color, background color or object instance, respectively) and keep everything else the same. We observe that on average a) on Novel Colors our model's performance dropped by 1.8% compared to 12.1% for CLIPort b) on novel backgrounds our model's performance dropped by 15.1% compared to 36.8% for CLIPort, with most of our failures caused by our low-level policy c) on novel object instances our model's performance dropped by 4.1% compared to 29.9% for CLIPort. We also attached [a graph of the performance comparison here](https://sites.google.com/view/spgem/home#h.ij4y30ojzm2x) in addition to detailed results in Table 6 and analysis in Section 6.2.1.
> - **Generalization tests on original CLIPort Benchmark**: We also show generalization experiments across novel colors and novel objects across the established CLIPort benchmark in Table 7 of the Appendix. Similar to above, our model generalizes better with novel objects and colors compared to CLIPort.
> - **Real World Experiments**: In our real-world experiments, we also use novel objects and novel object descriptions like "bananas", "strawberry", and "small objects" which the model hasn't seen during training (please refer to our [real world video executions](https://sites.google.com/view/spgem/home#h.g0v2pevzi5gc)).
>
> For more details and analysis of these results, please refer to Section 6.2.1 of our Appendix. While our work proposes a model for instruction-guided manipulation that generalizes better, we have by no means addressed this problem to its full generality and hence we also added appropriate qualifiers (like "better" generalize) in our introduction and conclusion sections.

---

> > ### Author Response · Authors · 2022-11-16
> > **Response to Reviewer hrSa (Part 2)**
> >
> >
> > >LLMPlanner such as PaLM-saycan uses success predictors to give some feedback whether goal state has been reached. Though such goal state detection through energy minimizer is discussed, I do not see whether such a feedback mechanism exist?
> >
> > We indeed did not have any such feedback mechanism in the paper. Inspired by your comment, **we added a success detection check** that works as follows: Once the model executes its self-generated goal, we re-detect all relevant objects using our VLM-grounder module and check if they are close to their predicted goal locations. We found that adding this goal detection helps a) in **knowing when the goal has not been achieved with a high precision of 76%** (please refer to Table 8 in the Appendix) b) and in fixing the errors by retrying which **leads to a 3.5% performance boost in composition-group benchmark and 1% boost in the composition-one-step benchmark** (please refer to the last row in Table 2 of main paper). We also visualize demonstrations with retrying based on goal detection feedback in section ["Simulation with Closed Feedback"](https://sites.google.com/view/spgem/home#h.bsv3bgk7x1mz).
> >
> > To the best of our knowledge, PaLM-saycan does not have success predictors, but another LLMPlanner "Inner Monologue" [1]. did add these feedback mechanisms over Say-Can. However, as they explain in Section A of their supplementary, their feedback mechanisms rely on ground truth information and/or hand-designed heuristics such as checking if the picked object has a 2D Euclidean distance <4cm from the place object, which is very narrowly applicable and would clearly fail for relations like left/right. Models that predict their own (visual) goals, like the model we propose, can easily add such feedback loops. We added a section on closed-loop feedback in our main paper, Section 3 and a detailed discussion on feedback in Section 6.2.2 in Appendix. Thank you for this great suggestion.
> >
> > [1]: Huang, Wenlong, et al. "Inner monologue: Embodied reasoning through planning with language models." arXiv preprint arXiv:2207.05608 (2022).
> >
> > ***
> >
> > > Some part has been discussed as physics based issues in Limitations. However, this is important, as the model only outputs final goal configuration; implying many goal configurations may be hard to achieve through pick-and-place policies in reality. What is the effect of this?
> > > Its easy to think of scenarios where the executor may not be able solve this using a greedy policy (stacks, tower of hanoi, inadvertently creating obstacles etc.).
> >
> >
> > Indeed a greedy pick and place policy will not be able to place the top block of a tower before the previous ones. However, this is not the limitation of pick-n-place policies but the lack of explicit learning of action order in our framework. As we mention in our limitations, our model makes the assumption that any ordering of objects is valid to achieve a specific scene goal configuration. And, indeed, as you are pointing out, greedily selecting this order can fail. One general solution is to add temporal energy priors which is a direct avenue for future work. We added a detailed discussion of this and other limitations in Section 6.3 of the Appendix.
> > ***
> >
> > >I am wondering is the set of binary and n-ary spatial concepts open? You at least need the knowledge of which is a binary vs n-ary predicate? How do you infer that in case you are not using a list of predicates? I see you are training one EBM per predicate, which means the predicate list should be fixed right?
> >
> > Yes, that is precisely right! The set of binary/n-ary concepts is not open and we need to know whether some concept is binary or n-ary. However, our EBM library is expandable – we can add new predicates and train  EBMs for each of them without re-training any component other than the language parser which needs to map the natural language with new predicates to the correct EBM graph. In contrast, baselines like CLIPort would need to be re-trained from scratch for each new task which involves learning perception and execution from scratch too.
> >
> > ***
> >
> > > Minor Comments
> > > More details on EBMs; algorithm block; details on scene-rearrangement; what is task progress in Table 1?
> >
> > Thank you for these suggestions. We added more details on EBMs in Section 6.1.2; an algorithm block for our pipeline in Algorithm-1 block, Section 6.1; details on scene-rearrangement benchmark creation in Section 6.5. Task-progress measures the fraction of subgoals completed by the model. For example, if the model achieves 4 out of 5 intended state changes, it gets a task progress reward of 0.8. We clarified this further in Section 4 of the main paper.
> >
> > ***
> >
> > Thank you again for your very constructive and concrete feedback and suggestions -- we believe these experiments have significantly strengthened our work.

---

> > > ### Comment · Reviewer_hrSa · 2022-11-23
> > > **Experiments, and Some More comments**
> > >
> > > Thank you for the experiments. Overall, the experiments done are quite impressive. But, in some cases, I feel the baselines are missing.
> > >
> > > Table 5, Real-world Experiments: While the results for the table-top experiment is interesting, I am wondering about what are the baseline performance you compared to. Can you report a baseline result such as using LLMPlanner or CLIPort?
> > >
> > > About binary/n-ary concepts: I am not fully convinced how easy or difficult it would be to extend it to external n-ary predicates. Can you give an example and explain?

---

> > > > ### Author Response · Authors · 2022-11-27
> > > > **Response to Experiments and Comments**
> > > >
> > > > Thank you for your response and questions. We address them below:
> > > >
> > > > > Table 5, Real-world Experiments: While the results for the table-top experiment is interesting, I am wondering about what are the baseline performance you compared to. Can you report a baseline result such as using LLMPlanner or CLIPort?
> > > >
> > > > Thank you for the suggestion, **we evaluated CLIPort and LLMPlanner in the real-world scenes**, and the results are below and on our [website](https://sites.google.com/view/spgem/#h.vra0b8l2me1u):
> > > >
> > > >
> > > > | Model      | Composition-one-step | Composition-group | Circles  | Lines    |
> > > > | ---------- | -------------------- | ----------------- | -------- | -------- |
> > > > | CLIPort    | 13.1                 | 22.9              | 34.0     | 46.0     |
> > > > | LLMPlanner | 39.5                 | 25.9              | 34.0        | 46.0        |
> > > > | Ours       | **85.6**             | **75.8**          | **94.0** | **90.0** |
> > > >
> > > >
> > > > We **significantly outperform both CLIPort and LLMPlanner in the real-world** experiments as well. Note that LLMPlanner cannot futher simplify shape concepts like "Make a circle" and hence LLMPlanner's performance is identical to CLIPort for "circles" and "lines".
> > > >
> > > > CLIPort did very poorly when applied in the real world scenes directly, with  performance close to 0. The numbers we report in the Table above are after performing background subtraction in the RGB input and depth map; only then we saw non-zero performance. The authors of CLIPort also report [here](https://github.com/cliport/cliport#real-robot-training-faq) that CLIPort might need 50-100 demos collected and annotated in real-world to achieve reasonable performance.
> > > > For their real-world experiments, the Inner Monologue paper [1] used a hand-designed execution policy (instead of CLIPort) which is only applicable when the place location is on top of some other object (like “put block on the plate”) and fails for tasks like “Place block to the left of the plate” where the place location is just a location on the table and not an object (Section 4.2 of [1]). That hand-design policy does not fit our purposes as it is too restrictive. For that reason, we evaluate the same version of LLM Planner we use in simulation (LLM+CLIPort), that is, using CLIPort as the execution short-term policy, helped by background subtraction.  Our method does not require neither background subtraction nor hand-designed execution policies. Unlike CLIPort, our model can utilize open-vocabulary detectors trained on real-world data. Unlike LLMPlanners which generate goals  in language form, we generate a visual goal which can be executed by a local execution policy without any hand-designing. Thus it is directly applicable to a real-world robot.
> > > >
> > > > We will modify Table-5 to include these baselines in the camera-ready version since we are not allowed to edit them on openreview presently.
> > > >
> > > > [1]: Huang, Wenlong, et al. "Inner monologue: Embodied reasoning through planning with language models." arXiv preprint arXiv:2207.05608 (2022).
> > > >
> > > > ***
> > > >
> > > > > About binary/n-ary concepts: I am not fully convinced how easy or difficult it would be to extend it to external n-ary predicates. Can you give an example and explain?
> > > >
> > > > Following your suggestion, we examined how we can extend our model for new n-ary predicates i.e. triangles and squares. All we need to do is 	a) train a separate object node EBM for triangles and squares, and b) extend the language parser to map utterances that contain triangle or square concepts to the corresponding EBMs. We do not need to change any other components i.e. the grounder and the low-level policy execution, since their task is agnostic to the spatial concepts in the instruction. This way, our concept EBM library can continually grow without changing any other components except the parser -- thus avoiding catastrophic forgetting.
> > > >
> > > > We show quantitative results for this extension below as well as on our [website](https://sites.google.com/view/spgem/#h.icfyx9oqt4a5). For our model, we only extend the EBM library with new EBMs and extend the parser. The grounder and the low-level execution policies are still trained on spatial-relations benchmark. (We also show CLIPort’s results as baseline – it is trained end-to-end from scratch on these demonstrations.)
> > > >
> > > >
> > > >
> > > > | Method  | Squares | Triangles |
> > > > | ------- | ------- | --------- |
> > > > | CLIPort | 32.2    | 43.9      |
> > > > | Ours        |    **94.1**   |    **96.1**    |
> > > >
> > > >
> > > > We look forward to your further feedback, please let us know if you have any other questions.

---

> > > > > ### Author Response · Authors · 2022-12-03
> > > > > **A gentle reminder for post-rebuttal discussion**
> > > > >
> > > > > Dear reviewer hrSa,
> > > > >
> > > > > Thank you for your effort in reviewing our paper. Following our discussion, we revised our paper accordingly to include
> > > > > - real-world experiments and extensive comparisons with the baselines
> > > > > - an exhaustive generalization analysis across different aspects (color, background, object classes)
> > > > > - comparisons on the original CLIPort Benchmark
> > > > > - closed-loop feedback
> > > > > - an illustration of the extensibility of our EBM library to solve new shape-based tasks
> > > > >
> > > > > Since few days are remaining for the discussion phase, we would appreciate it if you let us know whether you have any further questions.
> > > > >
> > > > > Sincerely,
> > > > >
> > > > > anonymous authors

---

### Author Response · Authors · 2022-11-16
**Summary of Revisions**

We thank all the reviewers (hrSa, MYGL, erLa) for their constructive and insightful feedback. The reviewers find our main idea interesting (hrSa, MYGL), definitely novel (hrSa), clearly written (hrSa, MYGL) with promising empirical results (MYGL, erLa) and fairly generalizable optimization scheme (erLa).

We have revised the paper and supplementary material on openreview. We also created a project website with various visual results here: [website](https://sites.google.com/view/spgem/). In addition to responses to specific reviewers, here we summarize the major revisions and additional experiments we have added to the paper:
- Real World Robot Experiments (hrSa, MYGL)
- Benchmarking Exhaustive Generalization (hrSa)
- Closed Loop Feedback (hrSa)
- Detailed Limitation Section with concrete future directions (MYGL)
- Robustness Experiments (MYGL)
- Baseline with hardcoded rules (erLa)
- EBM Library and Heatmap Visualizations (erLa)
- EBM Optimization visualization in simulation and real world (erLa)
- Pose-based EBMs (erLa)

These changes are marked in magenta in our revised paper and in the Appendix.

---

### Decision · Program_Chairs · 2023-01-20

**Decision:**

Reject

**Justification For Why Not Higher Score:**

This paper was on the decision frontier accept/reject.
It could have been accepted

**Justification For Why Not Lower Score:**

No lower score possible

**Metareview: Summary, Strengths And Weaknesses:**


This paper introduces an energy based layout model to perform spatial rearrangement actions conditioned on language, which can then be combined with a pick and place low-level policy in robotics tasks. The reviewers generally appreciated the idea of using energy based models in this context and to go from language to reasoning in spatial (bounding box) space.

The paper has received three expert reviews and was a tough nut to crack. The reviews were initially mixed and slightly unfavorable, and the paper actually stayed in the borderline area throughout the full evaluation period. It has been estimated to lie on the decision boundary by the reviewers and the AC and good arguments have been made for its rejection. In particular, several issues were raised:

- Positioning of the paper with respect to literature on layout models, in particular since existing work is more complex (in terms of constraints) compared to the submitted paper. This was the one remark where the authors' response is considered as hand-wavy.
- Missing baselines in comparisons
- Generalization capabilities
- Writing of the paper. This paper spends 4 pages on introduction and descriptions of related work and only a single page on the description of the method, which boils down to references of existing papers and a single equation. Details of the model are not given, the paper refers to the appendix, but even in the appendix details are omitted. This was considered as a major problem, as this paper is very far from being reproducible in its current form.

On the upside, the authors responded to (most) of the reviewers' comments and did a large number of new experiments, requested by the reviewers. While not all of them were considered convincing, they did address many of the raised points.

This paper has been extensively discussed in a meeting involving the AC and all three reviewers. No clear commitment has been made by the reviewers, who agreed that the paper had merit and was interesting to the community but also suffered from unfortunate choices made in presentation and evaluation. All reviewers agreed that there were shortcomings in evaluation, which had the potential to skew the perception of the problem and the field by a potential reader, but no consensus was reached on whether this was sufficient grounds for rejection.

The AC then discussed this paper with the SAC, and they considered all available information. Their message to the authors is very encouraging, and they thank them for their hard work during the rebuttal phase, providing additional experiments, which in the long run will be an important contribution to the paper. However, they also decided that in its current state the paper is not yet ready for publication. The points weighting most in this decision were the lack of positioning and evaluation (noted by all reviewers), but also the lack of reproducability. As it is written, the paper simply does not provide enough information to the reader on the model and its workings.

**Summary Of Ac-Reviewer Meeting:**

All said in the meta-review